# HiPRAG: Hierarchical Process Rewards for Efficient Agentic Retrieval Augmented Generation

**Peilin Wu[1], Mian Zhang[1], Kun Wan[2], Wentian Zhao[2], Kaiyu He[1], Xinya Du[1], Zhiyu Chen[1]**
[1]The University of Texas at Dallas, [2]Adobe Inc.
{peilin.wu,zhiyu.chen2}@utdallas.edu

## ABSTRACT

Agentic Retrieval-Augmented Generation (RAG) is a powerful technique for incorporating external information that Large Language Models (LLMs) lack, enabling better problem solving and question answering. However, suboptimal search behaviors exist widely, such as over-search (retrieving information already known) and under-search (failing to search when necessary), which leads to unnecessary overhead and unreliable outputs. Current training methods, which typically rely on outcome-based rewards in a Reinforcement Learning (RL) framework, lack the fine-grained control needed to address these inefficiencies. To overcome this, we introduce **Hi**erarchical **P**rocess Rewards for Efficient agentic **RAG** (HiPRAG), a novel training methodology that incorporates a fine-grained, knowledge-grounded process reward into the RL training. Our approach evaluates the necessity of each search decision on-the-fly by decomposing the agent's reasoning trajectory into discrete, parsable steps. We then apply a hierarchical reward function that provides an additional bonus based on the proportion of optimal search and non-search steps, on top of commonly used outcome and format rewards. Experiments on the Qwen2.5 and Llama-3.2 models across seven diverse QA benchmarks show that our method achieves average accuracies of 65.4% (3B) and 67.2% (7B), outperforming strong agentic RAG baselines. This is accomplished while dramatically improving search efficiency, reducing the over-search rate from over 27% in baselines from previous work to just 2.3% and concurrently lowering the under-search rate. These results demonstrate the efficacy of optimizing the reasoning process itself, not just the final outcome. Further experiments and analysis demonstrate that HiPRAG shows good generalizability across a wide range of RL algorithms, model families, sizes, and types. This work demonstrates the importance and potential of fine-grained control through RL, for improving the efficiency and optimality of reasoning for search agents. [1]

## 1 INTRODUCTION

Large Language Models (LLMs) augmented with retrieval have rapidly evolved into agentic RAG systems that can autonomously issue search queries, incorporate external knowledge, and perform multi-step reasoning (Singh et al., 2025; Li et al., 2025a; Zhang et al., 2025). In particular, recent frameworks integrate reinforcement learning (RL) to empower LLMs with the ability to decide when and what to retrieve during step-by-step reasoning (Jin et al., 2025b; Song et al., 2025a; Chen et al., 2025). However, along with this potential comes a critical drawback: today's agentic RAG agents often exhibit two suboptimal search behaviors: over-search, where the agent issues unnecessary or redundant retrievals (Wu et al., 2025; Qian et al., 2025b), and under-search, where it fails to retrieve external knowledge when it is actually needed, which undermine their accuracy and efficiency (Wu et al., 2025; Shen et al., 2024). These observations highlight that simply pairing LLMs with a search tool is not enough; the manner in which the agent uses the search tool must be optimized.

---

[1]We have released our code and model at `https://github.com/qualidea1217/HiPRAG`.

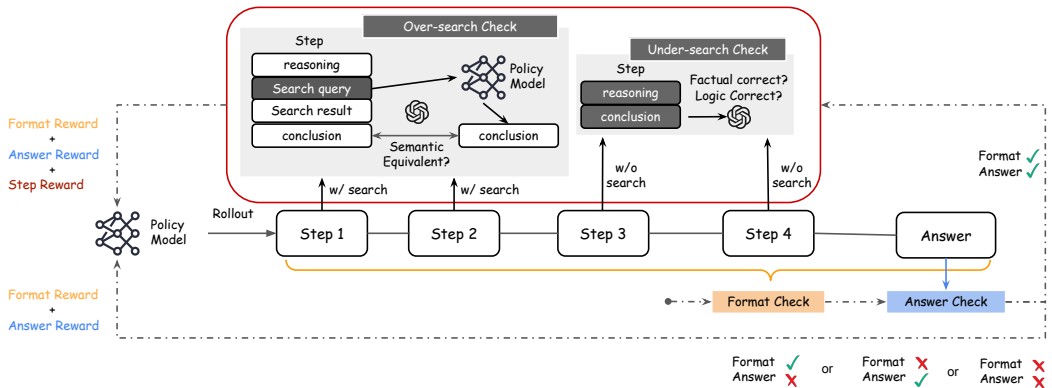

Figure 1: A general overview of the HiPRAG training workflow. The policy model generates a multi-step reasoning trajectory, and each step is evaluated on-the-fly to detect suboptimal search behaviors. A final hierarchical reward is then computed by combining a process bonus for step optimality with rewards for the final answer's correctness and proper formatting.

Recent research has turned to RL signals to tune the search behavior of agentic RAG system or search agents. One type of work proposed length or retrieval times-based penalties to encourage shorter reasoning trajectory (Song et al., 2025b; Wang et al., 2025a). Such heuristics can reduce redundant steps but risk oversimplifying the problem: the model may learn to avoid searches altogether even when searches are necessary (Wang et al., 2025a), therefore exacerbating under-search. Other researchers have incorporated model confidence (Wu et al., 2025) and knowledge awareness (Huang et al., 2025b; Sun et al., 2025) into the reward. These advances demonstrate the promise of process-level reward shaping in RAG. Yet, important limitations remain: confidence thresholds and knowledge classifiers are imperfect proxies that can misjudge when to search, and learned process reward models may introduce bias or only weakly align with true step quality. Crucially, none of these methods gives the agent explicit, step-specific feedback on each retrieval decision: whether a particular search superfluous or a missing search should have been taken is not enforced in a fine-grained way.

In this work, we introduce HiPRAG, a Hierarchical Process reward framework for agentic RAG, to address the above challenges. Instead of evaluating only by the final outcome or coarse proxies, HiPRAG explicitly parses the agent's reasoning trajectory into structured steps and constructs reward at multiple levels to optimize both correctness and efficiency. First, the agent is guided to produce a well-structured reasoning trajectory that is parsable by the rule-based method, with a correct final answer. On top of that, we design a process reward function that provides bonus reward that increases linearly as the ratio of optimal steps (not over-search or under-search) of the trajectory increases, accompanied by a set of fast and robust methods for detecting over-search and under-search for each step in the reasoning trajectory on-the-fly during training, achieved by directly prompting an external LLM for verification. These step-level bonus rewards are gated by the format and final answer's correctness to ensure that the agent is not over-punished for taking suboptimal steps if a correct format or answer is presented, which harms the reasoning ability. By hierarchically constructing the reward for format and outcome correctness as well as process quality, HiPRAG provides more fine-grained and accurate training signal than prior methods while avoiding the drawbacks of them.

We conduct experiments on seven question-answering benchmarks covering knowledge-intensive single and multi-hop queries. HiPRAG delivers significant gains in both accuracy and retrieval efficiency across diverse evaluations. Empirically, HiPRAG achieves average accuracies of 65.4% (3B) and 67.2% (7B), computed with Cover Exact Match, significantly outperforming strong agentic RAG baselines. More importantly, it achieves unprecedented gains in efficiency, reducing the over-search rate from over 27% in baselines from previous work (Wu et al., 2025) to just 2.3%, while concurrently lowering the under-search rate to as low as 29.0%. Our method shows good generalizability across different model families (Qwen2.5 and Llama-3.2), RL algorithms (PPO and

GRPO), model sizes (3B and 7B), and model types (base and instruct). Our major contributions are as follows:

- We propose HiPRAG, a novel Reinforcement Learning training methodology that uses a hierarchical, knowledge-aware process reward to provide fine-grained supervision on the search behavior of agentic RAG systems.
- We introduce an efficient and direct method for detecting over-search and under-search behaviors on-the-fly during training, accompanied with a set of output format parsable by rule-based methods, enabling the effective application of our process-based reward.
- We empirically demonstrate that our approach enhances both the accuracy and efficiency of various LLMs and RL algorithms, across multiple question-answering benchmarks, showing its strong generalizability.

## 2 RELATED WORK

**Agentic RAG & Tool Use**    The integration of external knowledge into LLMs has evolved into more dynamic, agentic systems. Frameworks like ReAct (Yao et al., 2023) demonstrated that LLMs can synergize reasoning and acting, paving the way for agents that can autonomously decide when and what to retrieve. This led to a new class of agentic RAG systems that interleave retrieval with multi-step reasoning, often structured as a "chain-of-thought" process (Trivedi et al., 2023). Variations like Chain-of-Retrieval (Wang et al., 2025b) and DeepRAG (Guan et al., 2025) further refined this by structuring the retrieval process itself into sequential steps to handle complex queries. To improve the decision-making capabilities of these agents, recent work has increasingly turned to RL (Singh et al., 2025; Li et al., 2025a; Zhang et al., 2025). RL frameworks train agents to develop optimal policies for invoking tools, including search engines. For instance, Huang et al. (2025a) use RL and curriculum learning to enhance RAG. This aligns with a broader trend of using RL to improve general tool use in LLMs. Works like ToolRL (Qian et al., 2025a) and ToRL (Li et al., 2025c) have shown that RL from rewards based on task success can significantly scale and improve the tool-integration capabilities of LLMs. Our work builds on this foundation, applying RL not just to achieve a correct final outcome but to optimize the efficiency of the retrieval process itself.

**Efficient Agentic RAG & Tool Use**    While agentic RAG enhances reasoning, it often introduces inefficiency, such as redundant or unnecessary tool calls. A key research direction has been to make retrieval adaptive, triggering it only when the model's internal knowledge is insufficient. Early approaches relied on heuristics or classifiers to detect uncertainty and determine the need for retrieval (Mallen et al., 2023; Dhole, 2025). More sophisticated methods learn to assess the LLM's real-time information needs or self-awareness from its internal states to make dynamic retrieval decisions (Su et al., 2024; Zubkova et al., 2025; Yao et al., 2025; Huanshuo et al., 2025). Concurrently, RL has emerged as a powerful tool for optimizing the efficiency of tool use. In the context of RAG, RL has been used to create search-efficient models (Sha et al., 2025) and to mitigate suboptimal search behaviors by reducing uncertainty (Wu et al., 2025). Works like R1-Searcher++ (Song et al., 2025b) and other synergistic reasoning agents (Huang et al., 2025b) use RL to incentivize dynamic and necessary knowledge acquisition. ReARTeR (Sun et al., 2025) introduces a framework with a trustworthy process reward model to score and refine each step in a RAG pipeline. This mirrors efforts in the broader tool-use domain to mitigate tool overuse. For example, SMART (Qian et al., 2025b), SMARTCAL (Shen et al., 2024), and OTC (Wang et al., 2025a) train agents to be self-aware and make optimal tool calls, often using RL. Yue et al. (2025) and Ye et al. (2025) also use verifiable stepwise rewards to promote more efficient general reasoning paths. While these methods improve efficiency, they often rely on proxies like model confidence or separately trained reward models. HiPRAG differs by introducing a direct, on-the-fly evaluation of each search step's necessity, providing a more explicit training signal for efficiency.

## 3 HIPRAG

Our approach, which we term HiPRAG, introduces a fine-grained process-based reward mechanism into the reinforcement learning loop for training agentic RAG systems. This is achieved through three key aspects: (1) A redesigned, explicitly structured output format that enables rule-based

parsing of reasoning steps described in Section 3.1; (2) An efficient method for detecting over-search and under-search behaviors described in Section 3.2; and (3) A hierarchical reward function that dynamically prioritizes correctness and search efficiency described in Section 3.3. An overview of the HiPRAG workflow can be found in Figure 1.

## 3.1 DECOMPOSING REASONING TRAJECTORY INTO PARSABLE STEPS

A primary obstacle to implementing process rewards in agentic RAG is the difficulty of parsing an agent's reasoning trajectory, which includes one or more reasoning steps. For each step, the agentic RAG system resolves one sub-query by either using search tools or using its own parametric knowledge. Existing frameworks like Search-R1 (Jin et al., 2025b) often generate reasoning within a series of `<think>` XML blocks, interleaved with other blocks about search queries and retrieved information. This format, though maintaining the fluency of reasoning, makes it hard to isolate and evaluate individual steps of reasoning for the following two reasons: (1) **Ambiguous Step Boundaries**: A single `<think>` block does not map to a discrete reasoning step. It often mixes the conclusion from a previous action with the reasoning and planning for the current action, making it difficult to programmatically isolate a self-contained step. (2) **Implicit Internal Reasoning**: Non-search steps, where the agent relies on its parametric knowledge, are not explicitly tagged. They are embedded as prose within the `<think>` block, making it challenging to differentiate them from the analytical text that precedes a search query without extra natural language understanding. Therefore, this format makes it infeasible to isolate and evaluate individual steps of reasoning during RL training, as it would require slow and expensive calls to a powerful LLM for post-hoc interpretation, which significantly slows down the RL training process and introduces a high error rate.

To overcome this, we enforce a structured, machine-parsable output format during RL training. We modify the agent's prompt and rollout logic to generate its entire reasoning trajectory within a single `<think>` block, which in turn contains a sequence of discrete `<step>` blocks. Each step can either be a search step or a non-search step, distinguished by the presence of a `<search>` block containing search queries and `<context>` block containing retrieved information. Formally, a complete reasoning trajectory $T$ for a given question is a sequence of $n$ steps with a final answer $a$, $T = \{s_1, s_2, ..., s_n, a\}$. Each step $s_i$, for $i \in [1, n]$ can either be a search step, which is a tuple $s_i^R = (r_i, q_i, c_i, o_i)$ if the model chooses to search, or be a non-search step, which is a tuple $s_i^{NR} = (r_i, o_i)$, where $r_i$ is the model's reasoning block containing the planning and analysis within this step, $q_i$ is the search query, $c_i$ is the retrieved context, $o_i$ is the conclusion or summary of the knowledge gained in the current step. An example of a pipeline that does the inference for this format is shown in Algorithm 1. An example comparing format before and after for the same question with the equivalent reasoning trajectory is included in Figure 3.

We ensure adherence to this schema through two approaches in parallel. First, the agent's system prompt is updated with explicit instructions and few-shot examples demonstrating the correct usage of all the XML tags. Figure 4 shows the prompt with parsable output format. Second, as detailed in Section 3.3, our RL framework applies a positive reward for correct outputs, thus incentivizing the model to consistently produce parsable trajectories.

## 3.2 ON-THE-FLY DETECTION OF SUBOPTIMAL SEARCHES

With trajectories segmented into discrete steps, we can implement efficient checks for over-search and under-search during the RL training phase.

**Over-search Detection.** Previous methods for detecting over-search involved complex re-generation pipelines, where the search context was removed and a fixed instruction was appended to prompt the model to rely on its internal knowledge (Wu et al., 2025). This approach is brittle, as the appended instruction can conflict with the agent's original reasoning flow and produce unnatural, low-quality outputs. It is also computationally expensive to re-generate if the search appears at the end of a long reasoning trajectory. We propose a more direct and robust method. For each search step $s_i^R = (r_i, q_i, c_i, o_i)$, we take its search query $q_i$ and directly prompt the policy model with it as a standalone question. We then obtain a re-generated answer, $o_i'$. An external LLM judge is used to assess the semantic equivalence of the original step's conclusion $o_i$ and the re-generated answer $o_i'$. The prompt for external LLM judge is shown in Figure 5. If they are equivalent, the search was

redundant, and the step is flagged as an over-search. This method is not only faster but also provides a more reliable signal by isolating the core knowledge required by the query.

**Under-search Detection.** For each non-search step $s_i^{NR} = (r_i, o_i)$, we verify the factual and logical accuracy of its reasoning $r_i$ and conclusion $o_i$ by prompting an external verifier model to assess the correctness of $r_i$ and $o_i$. The prompt for external verifier model is shown in Figure 6. If the content is found to be incorrect, the step is flagged as an under-search, as the agent failed to utilize the search tool to retrieve necessary information, leading to a hallucination or factual error. Some under-search cases may appear as suboptimal or incomplete but locally correct intermediate steps. We intentionally avoid penalizing these cases because step completeness often depends on the agent's global multi-step plan and is subjective to determine at the step level without expensive counterfactual supervision. Penalizing correct-but-incomplete steps tends to push agents toward over-search by discouraging reliance on parametric knowledge.

In actual implementation, both of the detection methods can work concurrently to improve the detection speed. For conducting over-search detection over a batch of data during RL rollout phase, the re-generation step can be executed separately through batch generation before using the external LLM judge to further improve the training speed.

## 3.3 HIERARCHICAL PROCESS REWARD CALCULATION

A naive length or confidence-based reward function that penalizes search can over-suppress the agent's retrieval capabilities, leading to poor performance on knowledge-intensive tasks. Our goal is to incentivize optimal search behavior to improve performance and efficiency, while maintaining the basic ability of reasoning with a search tool. In order to achieve this, the rewards need to be dynamically focused on incentivizing format and final answer correctness at the early stage of the RL training, while shifting its focus towards reasoning efficiency and optimality after the basic search ability has been established, by providing a higher reward to be differentiated from the original outcome and format reward. To this end, we design a hierarchical reward function that prioritizes correctness and format adherence before rewarding process optimality, while shift its focus towards process optimality after the reasoning ability has established, on top of the outcome + format reward version of Search-R1 reward (Jin et al., 2025a).

**Hierarchical process reward.** Let $A(T) \in \{0, 1\}$ indicate the final answer $a$ of the trajectory $T$'s correctness (here we use Cover Exact Match as introduced in Section 4.1), and $F(T) \in \{0, 1\}$ indicate that the trajectory follows the required format (an example of the implementation of $F(T)$ can be found in Algorithm 2 and 3). Let $N(T)$ be the number of steps in the trajectory $T$, with the number of optimal (neither over-search nor under-search) steps $N_{\text{corr}}(T)$ be

$$N_{\text{corr}}(T) = \left| \left\{ s^R \in (T) : \neg\mathsf{Over}(s^R) \right\} \right| + \left| \left\{ s^{NR} \in (T) : \neg\mathsf{Under}(s^{NR}) \right\} \right|$$

where $\mathsf{Over}(\cdot)$ and $\mathsf{Under}(\cdot)$ are the detectors from Section 3.2. With a format weight $\lambda_f \in [0, 1]$ and a process bonus coefficient $\lambda_p \geq 0$, we define a single merged reward:

$$R(T) = A(T)\big(1 - \lambda_f\big) + \lambda_f F(T) + \lambda_p A(T)F(T) \frac{N_{\text{corr}}(T)}{N(T)}. \tag{1}$$

This expression is algebraically equivalent to the standard outcome + format reward used in prior work when $\lambda_p{=}0$, and it adds a gated process bonus only when both the answer and the format are correct. In particular, the reward becomes $R(T) = 1 + \lambda_p \frac{N_{\text{corr}}(T)}{N(T)}$ whenever $A(T){=}F(T){=}1$. A collection of all the symbols used can be found in Table 3.

This hierarchical structure ensures that the agent is first incentivized to produce well-formed reasoning trajectory with correct answers. Only once it achieves this primary goal does it receive an additional reward bonus for the efficiency and validity of its reasoning path, which avoids the pitfalls of over-suppression while directly encouraging the model to develop a more nuanced understanding of its own knowledge boundaries.

## 4 EXPERIMENT SETUP

This section details the experimental framework used to evaluate HiPRAG. We outline the datasets, evaluation metrics, models, and training procedures to ensure reproducibility and provide a clear context for our results.

### 4.1 DATASETS & METRIC

Our experimental data is composed of a wide coverage of both single and multi-hop Question Answering (QA) samples, modeled after the one used in Search-R1 to ensure a fair and direct comparison with prior work. The training set is a combination of the official training sets from NQ (Kwiatkowski et al., 2019) and HotpotQA (Yang et al., 2018). This creates a diverse corpus for teaching the agent both single-fact retrieval and multi-hop reasoning, which is crucial for learning efficient reasoning. To assess both in-domain and out-of-domain generalization, we evaluate our models on a comprehensive test set composed from the development or test set of seven QA datasets: NQ, PopQA (Mallen et al., 2023), TriviaQA (Joshi et al., 2017), 2WikiMultiHopQA (Ho et al., 2020), Bamboogle (Press et al., 2023), HotpotQA, and MuSiQue (Trivedi et al., 2022).

The primary metric for our evaluation is Cover Exact Match (CEM) (Song et al., 2025a). This metric determines correctness by checking if the ground-truth answer string is present in the model's generated answer. We choose CEM over a strict Exact Match because modern LLMs are optimized for generating longer, explanatory responses. Imposing a strict match would penalize valid answers embedded in more verbose text and might not accurately reflect the model's capabilities. We also assess the model's efficiency by measuring the Over-search Rate (OSR) and Under-search Rate (USR), which are defined as the ratio of over-search and under-search steps among all identifiable search steps and non-search steps, within a set of test samples. Given a set of samples $\mathcal{D}_{\text{test}}$ with their reasoning trajectory $T$, the OSR and USR can be calculated according to Equation 2.

$$\text{OSR} = \frac{\sum_{T \in \mathcal{D}_{\text{test}}} |\{s^R \in T : \text{Over}(s^R)\}|}{\sum_{T \in \mathcal{D}_{\text{test}}} |\{s^R \in T\}|}, \quad \text{USR} = \frac{\sum_{T \in \mathcal{D}_{\text{test}}} |\{s^{NR} \in T : \text{Under}(s^{NR})\}|}{\sum_{T \in \mathcal{D}_{\text{test}}} |\{s^{NR} \in T\}|} \quad (2)$$

### 4.2 BASELINES

We compare our method against a comprehensive set of baselines that represent different paradigms in retrieval-augmented generation: (1) **Direct Inference**: Direct generation without any retrieval mechanism. (2) **Standard RAG**: A conventional RAG setup where retrieval is performed once based on the initial query (Lewis et al., 2020). (3) **Prompt-Based Agentic RAG**: Methods that rely on sophisticated prompting to achieve multi-step reasoning and search, including IRCoT (Trivedi et al., 2023) and Search-o1 (Li et al., 2025b). (4) **RL-Based Agentic RAG**: State-of-the-art methods that use reinforcement learning to train search agents, including Search-R1 (Jin et al., 2025b), R1-Searcher (Song et al., 2025a), R1-Searcher++ (Song et al., 2025b), and $\beta$-GRPO (Wu et al., 2025). Among these, R1-Searcher++ and $\beta$-GRPO are explicitly designed to improve search efficiency, making them strong baselines focused on efficiency.

### 4.3 TRAINING DETAILS

All RL-based models were trained with four NVIDIA A100 80GB GPUs. The training process was conducted for a total of 400 steps and saved the checkpoint every 50 steps. For evaluation, we adopted the following checkpointing strategy: if the training process completed without instability, the final saved checkpoint is used for testing. However, if the training reward collapsed, we use the last stable checkpoint saved before the collapse to ensure a fair evaluation of the model's best-learned state.

Our experiments leverage several models for different roles within the framework. The main experiments are conducted using the Qwen2.5-(3B/7B)-Instruct models (Qwen et al., 2025). To analyze performance across different model families and types, we also conduct experiments with Llama-3.2-3B-Instruct (Grattafiori et al., 2024) and Qwen2.5-3B. For the detection of suboptimal searches during training and evaluation, we utilize small-sized proprietary models that provide fast inference speed while maintaining sufficient performance. Over-search detection is performed by GPT-4.1 mini (OpenAI, 2025a), while under-search detection relies on GPT-5 mini (OpenAI, 2025b).

| Method | General QA | | | Multi-Hop QA | | | | Avg. |
|---|---|---|---|---|---|---|---|---|
| | NQ | TriviaQA | PopQA | HotpotQA | 2Wiki | MuSiQue | Bamboogle | |
| Direct Inference | 27.0 | 26.8 | 40.1 | 58.7 | 16.0 | 7.9 | 15.9 | 31.8 |
| Standard RAG | 51.2 | 54.7 | 65.7 | 56.9 | 21.6 | 18.5 | 18.6 | 45.3 |
| IRCoT | 27.5 | 36.0 | 42.5 | 51.4 | 37.6 | 19.4 | 20.6 | 36.4 |
| Search-o1 | 40.2 | 42.2 | 58.5 | 56.1 | 45.6 | 15.1 | 19.3 | 43.9 |
| R1-Searcher | 60.0 | 73.0 | 58.2 | 60.4 | 60.3 | 32.9 | 55.8 | 60.6 |
| R1-Searcher++ | 61.0 | 73.5 | 59.0 | **64.2** | 63.2 | 32.3 | **58.7** | 62.1 |
| Search-R1 | 61.2 | 73.6 | 56.5 | 54.0 | 63.6 | 24.8 | 48.4 | 60.3 |
| Search-R1-step* | 62.4 | 74.4 | 57.3 | 54.8 | 64.2 | 25.3 | 49.6 | 61.2 |
| $\beta$-GRPO | 65.0 | 75.0 | 60.0 | 53.0 | 66.0 | 24.0 | 52.0 | 62.5 |
| $\beta$-GRPO-step* | 62.4 | 73.9 | 61.3 | 52.1 | 66.0 | 22.8 | 54.4 | 62.1 |
| **HiPRAG-3B** | 68.7 | 75.5 | **66.3** | 57.4 | 67.4 | 24.1 | 41.6 | 65.4 |
| **HiPRAG-7B** | **71.2** | **76.3** | 63.2 | 62.4 | **71.7** | **34.1** | 52.8 | **67.2** |

Table 1: Main results of CEM (higher is better) in percentage on seven QA benchmarks. Best and second-best *overall* averages are marked in **bold** and underline. Search-R1-step* refers to the model trained with Search-R1 v0.3's output + format reward with HiPRAG's output format. $\beta$-GRPO-step* refers to the model trained with $\beta$-GRPO's reward with HiPRAG's output format. The HiPRAG-3B and 7B here refer to the model with highest Avg. CEM score among all trained HiPRAG models.

For our main experiments, we use Proximal Policy Optimization (PPO) (Schulman et al., 2017) as the core RL algorithm. PPO is selected for its demonstrated training stability in complex LLM fine-tuning scenarios, especially on the previous works on search agent development using RL. To assess the impact of the RL algorithm choice, we also perform experiments using Group Relative Policy Optimization (GRPO) (Shao et al., 2024). For GRPO we keep the same training parameters as the PPO training with a group size of 5.

For our retrieval environment of all experiments, we follow the Search-R1 setup and use the 2018 Wikipedia dump (Karpukhin et al., 2020) as the knowledge source, with E5-base (Wang et al., 2024) serving as the retriever. In each search step, the top-3 relevant passages are returned.

In terms of the inference parameters, we set the temperature and top p to 1 during the rollout stage of RL training to make sure a high possibility of generating desired reasoning trajectory. In testing, we set the temperature and top p to the models' default value. For hyperparameters in the reward function, we set $\lambda_f$=0.2 and $\lambda_p$=0.4 for the main experiments. We also explore the results under different $\lambda_p$ and $\lambda_f$ in Section 5.3 and Section 5.2.

## 5 RESULTS & ANALYSIS

This section presents a comprehensive analysis of HiPRAG. We evaluate its performance against state-of-the-art baselines and conduct detailed studies on the influence of model size, model family, and reinforcement learning algorithms. We also include ablation studies to validate our design choices and conclude with a qualitative case study.

### 5.1 MAIN RESULTS

We compare HiPRAG against a suite of strong baselines on seven question-answering benchmarks. As shown in Table 1, our approach outperforms all baseline methods across both 3B and 7B model sizes. The HiPRAG-7B model achieves an Avg. CEM score of 67.2%, a notable improvement over the next-best baseline, R1-Searcher++ (62.1%). This demonstrates that our fine-grained, process-based reward mechanism effectively guides the agent to develop more robust and accurate reasoning trajectories. We also present a qualitative case study comparing the reasoning trajectory from baseline and HiPRAG-trained model in Appendix K, as well as a brief analysis of efficacy of our methods in Appendix F. We also report the raw average number of retrieval calls per question (Avg. #Searches). Under a controlled Qwen2.5-3B-Instruct + PPO setup, HiPRAG reduces Avg. #Searches from 2.45 (Search-R1) and 2.15 ($\beta$-GRPO) down to 1.75 on average across seven benchmarks, corresponding to a 29% and 19% reduction respectively (Table 12).

| Base Model | RL Algo. | Method | Avg. CEM | Avg. OSR | Avg. USR |
|---|---|---|---|---|---|
| Llama-3.2-3B-Instruct | PPO | baseline | 56.4 | 7.3 | 57.6 |
| Llama-3.2-3B-Instruct | PPO | HiPRAG | 64.8 | 6.0 | 49.7 |
| Qwen2.5-3B-Instruct | GRPO | baseline | 58.5 | 8.4 | 52.1 |
| Qwen2.5-3B-Instruct | GRPO | HiPRAG | 64.4 | 4.1 | 33.2 |
| Qwen2.5-3B | PPO | baseline | 60.3 | 3.8 | 44.0 |
| Qwen2.5-3B | PPO | HiPRAG | 65.4 | 3.2 | 41.9 |
| Qwen2.5-3B-Instruct | PPO | baseline | 59.3 | 6.1 | 47.5 |
| Qwen2.5-3B-Instruct | PPO | HiPRAG | 64.1 | 4.9 | 38.1 |
| Qwen2.5-7B-Instruct | GRPO | baseline | 61.2 | 5.2 | 43.3 |
| Qwen2.5-7B-Instruct | GRPO | HiPRAG | **67.2** | **2.3** | 32.6 |
| Qwen2.5-7B-Instruct | PPO | baseline | 53.3 | 7.6 | 33.9 |
| Qwen2.5-7B-Instruct | PPO | HiPRAG | 64.5 | 6.2 | 29.0 |
| Qwen2.5-3B-Instruct | PPO | HiPRAG (over-search only) | 58.8 | 4.9 | 52.7 |
| Qwen2.5-3B-Instruct | PPO | HiPRAG (under-search only) | 63.3 | 6.6 | **16.9** |
| Qwen2.5-3B-Instruct | PPO | HiPRAG ($\lambda_p = 0.2$) | 59.6 | 5.5 | 44.5 |
| Qwen2.5-3B-Instruct | PPO | HiPRAG ($\lambda_p = 0.6$) | 62.5 | 5.2 | 39.0 |

Table 2: Summary of average scores in percentage of CEM (higher is better), OSR (lower is better), and USR (lower is better) for HiPRAG models trained with different parameters. Best and second-best values for each metric are marked in **bold** and underline. Baseline here refers to the reward with process bonus disabled ($\lambda_p = 0$). A detailed report on each individual dataset is in Appendix E.

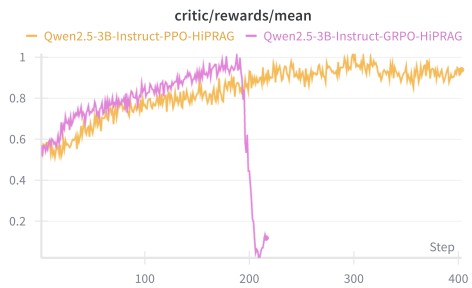

(a) Reward curves of Qwen2.5-3B-Instruct model trained with PPO/GRPO + HiPRAG with respect of the training steps.

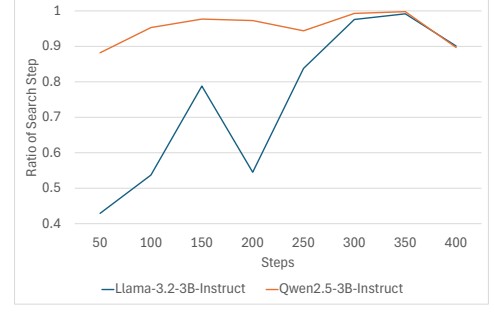

(b) Curves of the ratio of searches among all reasoning steps of Qwen2.5-3B-Instruct and Llama-3.2-3B-Instruct.

Figure 2: Reward curves for different RL algorithm and curves of the ratio of searches among all reasoning steps for different model families.

## 5.2 ANALYSIS ON INDIVIDUAL PARAMETERS

**Influence of Model Size** Larger models generally exhibit stronger reasoning capabilities with better accuracy. Our experiments confirm this trend, according to Table 2, as HiPRAG-trained 7B models consistently outperform their 3B counterparts. However, our process-based reward approach allows smaller models to achieve remarkable performance, closing the gap on larger models. For instance, our HiPRAG model trained based on Qwen2.5-3B-Instruct + GRPO (64.4% Avg. CEM) not only surpasses strong external 7B baselines like R1-Searcher++ (62.1% Avg. CEM), but it also outperforms the 7B counterpart trained with our baseline reward (61.2% Avg. CEM). This indicates that our training methodology is a more effective path to performance gains than simply scaling the model size with conventional rewards. Furthermore, a larger model size tends to provide more efficient search decisions. This is evident with GRPO, where the 7B model, on top of its higher accuracy, exhibits better efficiency (2.3% Avg. OSR, 32.6% Avg. USR) than the 3B model (4.1% Avg. OSR, 33.2% Avg. USR).

**Influence of Model Family** To assess the generalizability of HiPRAG, we trained both the Qwen2.5-3B-Instruct and Llama-3.2-3B-Instruct models with HiPRAG for comparison. Although both models achieve comparable peak accuracy after training with HiPRAG, their underlying be-

havior and efficiency differ. As shown in Figure 2b, the Llama-3B model shows a higher tendency to rely on its parametric knowledge with more non-search steps initially, resulting in a higher under-search rate. After training, the Qwen-3B model achieves its high Avg. CEM of 64.1% with lower suboptimal search rates (4.9% Avg. OSR, 38.1% Avg. USR) compared to the Llama-3B model's 6.0% Avg. OSR and 49.7% Avg. USR, based on Table 2. This suggests that while HiPRAG is effective across different model families, the base model's inherent tendencies can influence final search efficiency.

**Influence of RL Algorithm**    To explore the impact of different RL algorithms on HiPRAG method, we experimented with both PPO and GRPO on Qwen2.5-3B/7B-Instruct models. PPO offers better training stability, often completing the full training run without reward collapse, whereas GRPO consistently has the potential to achieve higher final performance and converges faster. A comparison of reward curve for Qwen-3B models can be found in Figure 2a. As seen in Table 2, GRPO yields a higher Avg. CEM for both 3B (64.4% vs. 64.1%) and 7B (67.2% vs. 64.5%) models and results in more efficient search behavior (e.g., 2.3% OSR for 7B-GRPO vs. 6.2% for 7B-PPO). This aligns with findings in related literature, where GRPO's critic-free approach often proves to be more sample-efficient for LLM training with the trade-off of less training stability.

**Influence of Format Reward Weight**    We experimented with different format reward weights $\lambda_f \in \{0.2, 0.4, 0.6\}$ while keeping $\lambda_p=0.4$ fixed (Qwen2.5-3B-Instruct + PPO). $\lambda_f=0.2$, used in our main experiments, provides the best trade-off: it achieves the highest Avg. CEM (64.1%) and the best Avg. OSR (4.9%), while larger values over-emphasize the instrumental goal of format correctness and degrade performance (Table 14).

**Influence of Instruction Tuning on Base Model**    To understand the impact of instruction-tuning on the base model before applying HiPRAG for RL, we compared the performance of HiPRAG on a base model (Qwen2.5-3B) and its instruction-tuned counterpart (Qwen2.5-3B-Instruct). Based on Table 2, the instruct-tuned model exhibited a higher initial reward, as its pre-training makes it more adept at following the structured output format required by our framework. Our hierarchical reward, which gates the process bonus until both the answer and format are correct, favors models that quickly learn this structure. However, the base model eventually caught up, converging to a similar reward level. Interestingly, the base model, once fully trained, may achieve higher Avg. CEM score (65.4% vs. 64.1%) and lower Avg. OSR (3.2% vs. 4.9%). This could be because it learns the reasoning and search behaviors from the RL objective more purely, without the potential biases introduced during the instruction-tuning phase.

## 5.3    ABLATION STUDIES

To validate the key components of the HiPRAG methodology and systematically isolate the sources of its performance gains, we conducted a series of ablation studies. These experiments are designed to deconstruct our approach by: (1) evaluating the impact of the new parsable output format independent of the process reward, (2) determining the optimal weighting of the process bonus coefficient, $\lambda_p$, which governs the reward hierarchy, and (3) demonstrating the necessity of addressing both over-search and under-search behaviors concurrently, rather than in isolation.

**Influence on Output Format**    To isolate the effect of the format and reward change, we trained a model variant called Search-R1-step*. This model uses the same outcome + format reward as the original Search-R1 v0.3 model (Jin et al., 2025a) but is enforced to use our new parsable output format. Besides, we also adapt the format change to $\beta$-GRPO and trained $\beta$-GRPO-step*. The results from both variants in Table 1 show that our structured format maintains the performance, with slight increase in some of the datasets. This confirms that the new parsable output format is a robust foundation and that the significant performance gains of our full method are attributable to the process-based reward mechanism it enables, not merely an artifact of the format change.

**Influence of Process Bonus Coefficient**    We tested our hierarchical reward with different values for the process bonus coefficient $\lambda_p$, which determines the weight of the step-correctness ratio. Based on Table 2, a coefficient of 0.4 provided the optimal balance, yielding the highest performance (64.1% Avg. CEM). A lower value of 0.2 behaved similarly to an outcome-only reward, failing to

sufficiently incentivize efficiency (59.6% Avg. CEM). This is reflected in its higher Avg. OSR of 5.5% and Avg. USR of 44.5%. A higher value of 0.6 over-prioritized step purity at the expense of final answer correctness, leading to a slight performance degradation (62.5% Avg. CEM). The optimal $\lambda_p$ of 0.4 achieved the best trade-off, with a low 4.9% Avg. OSR and 38.1% Avg. USR. Importantly, $\lambda_p$ controls the relative reward spread among already-correct trajectories: when $\lambda_p$ is too large, the gradient becomes dominated by differences in step ratios among correct outputs, encouraging short, conservative plans and reducing the model's willingness to add corrective steps that could fix borderline errors.

**Training with Over-search or Under-search Only**   We isolated the components of our process reward by training models with penalties for only over-search or only under-search. In these ablations, the ignored step type is *excluded* from the computation of process bonus ratio. If a trajectory contains zero steps of the evaluated type, we set the process bonus to the average bonus within the current training batch for stability. Based on Table 2, training to reduce only over-search proved insufficient, yielding a low Avg. CEM of 58.8%. While this approach successfully reduced the Avg. OSR to 4.9%, it caused the model to become too hesitant to search, resulting in a very high Avg. USR of 52.7%. Targeting only under-search was more effective (63.3% Avg. CEM), underscoring that preventing hallucination is more critical than improving efficiency. This method dramatically lowered the Avg. USR to just 16.9% but made the agent overly reliant on its search tool, slightly increasing the Avg. OSR to 6.6%. However, the best performance was achieved only when penalizing both suboptimal behaviors simultaneously (64.1% Avg. CEM), confirming that a holistic approach to search optimization is necessary.

## 6   CONCLUSION

This work tackles the pervasive inefficiencies in agentic RAG systems, which are often trained with coarse, outcome-based rewards that fail to correct suboptimal search behaviors. We introduced HiPRAG, a novel training methodology that incorporates a hierarchical, knowledge-aware process reward to instill more efficient reasoning. By decomposing agent trajectories into parsable steps and evaluating the necessity of each search action on-the-fly, HiPRAG provides the fine-grained supervision necessary to curb both over-searching and under-searching. Our experiments show that this approach not only achieves state-of-the-art performance on 3B and 7B models but also dramatically improves efficiency, reducing the over-search rate and under-search rate. This research validates that the path to creating powerful and efficient LLM search agents lies in optimizing the reasoning process itself, not just the final outcome.

## ETHICS STATEMENT

The authors of this paper have read and adhered to the ICLR Code of Ethics. This work, focuses on improving the efficiency and accuracy of AI agents, and we have considered the ethical implications of our methodology. While any capable AI system presents potential for misuse, our method aims to foster more reliable and grounded AI by explicitly penalizing factual errors (under-searching) and reducing computational waste (over-searching). We acknowledge that the system may inherit biases from its underlying data (Wikipedia) and foundation models. By improving agent efficiency, our work also contributes to more sustainable AI by reducing the energy consumption of deployed models.

## REPRODUCIBILITY STATEMENT

We are committed to ensuring the reproducibility of our work. Section 4 provides a comprehensive overview of our experimental framework, including the specific training and evaluation datasets, evaluation metrics (Cover Exact Match, Over-search Rate, and Under-search Rate), and baselines used for comparison. Section 4.3 details our training procedure, specifying the models (Qwen2.5, Llama-3.2), RL algorithms (PPO, GRPO), hardware, retrieval environment, and key hyperparameters. The core components of our HiPRAG methodology-the structured output format, the on-the-fly detection mechanism, and the hierarchical reward function-are thoroughly described in Section 3. To further facilitate replication, the Appendix contains the pseudocode for our inference process

(Algorithm 1) , along with the exact prompts for enforcing the parsable output format (Figure 4) and for detecting suboptimal searches (Figures 5 and 6).

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

APPENDIX

## A    EXAMPLE OF SEARCH-R1 FORMAT VS. HIPRAG FORMAT

In this section we present an example of transforming from the original Search-R1 output format to our HiPRAG output format shown in Figure 3. Original format interleaves multiple `<think>` blocks with search operations, making step boundaries ambiguous. Our format uses explicit `<step>` tags within a single `<think>` block. Each colored region represents a logical reasoning step. Our format clearly delineates these with `<step>` tags, enabling deterministic parsing. Step 2 (orange) demonstrates internal reasoning without search. The original format embeds this implicitly in prose; our format explicitly marks it as a step with only `<reasoning>` and `<conclusion>` tags.

## Comparison of Reasoning Trajectory Formats

| (a) Original Search-R1 Format | (b) HiPRAG Parsable Format |
|---|---|
| **Question:** Who is the current CEO of the company that makes the graphics cards used in the latest PlayStation, and what was their stock price at the close of the last trading day? | **Question:** Who is the current CEO of the company that makes the graphics cards used in the latest PlayStation, and what was their stock price at the close of the last trading day? |

(a) Original Search-R1 Format

```
<think>
Okay, this is a multi-part question.
I need to figure out the latest
PlayStation, who makes its graphics
card, who the CEO of that company
is, and their recent stock price.
I'll start by finding the most recent
PlayStation console.
</think>
<search> latest playstation console
model </search>
<information>
The latest PlayStation console is
the PlayStation 5 (PS5), which was
released by Sony in November 2020.  It
features a custom AMD RDNA 2-based
GPU.
</information>
<think>
So, the latest console is the
PlayStation 5,

and the graphics card is made by AMD.

Now that I've identified the company
as AMD, I need to find its current
CEO.
</think>
<search> current CEO of AMD </search>
<information>
Dr.  Lisa Su is the chair and chief
executive officer of Advanced Micro
Devices (AMD). She has held the CEO
position since 2014.
</information>
<think>
Great, the CEO of AMD is Dr.  Lisa Su.

The final piece of information I need
is the closing stock price from the
last trading day.  Since today is
Saturday, July 19, 2025, I need the
price from Friday, July 18, 2025.
</think>
<search> AMD stock price closing July
18 2025 </search>
<information>
On Friday, July 18, 2025, the stock
for Advanced Micro Devices, Inc.
(ticker:  AMD) closed at a price of
$175.40 per share.
</information>
<think>
I have all the necessary information.
The company is AMD, the CEO is Dr.
Lisa Su, and the closing stock price
was $175.40.  I can now provide the
final answer.
</think>
<answer> Dr.  Lisa Su and $175.40.
</answer>
```

(b) HiPRAG Parsable Format

```
<think>
<step>
<reasoning> This is a multi-part
question.  First, I need to identify
the latest PlayStation console.  Then,
I need to find out which company
makes its graphics cards.  I suspect
it's either Sony itself, NVIDIA, or
AMD, but I need to be certain.   I
will start by identifying the latest
PlayStation model. </reasoning>
<search> latest playstation console
model </search>
<context> The latest PlayStation
console is the PlayStation 5 (PS5).
</context>
<conclusion> PlayStation 5 (PS5)
</conclusion>
</step>
<step>
<reasoning> Now that I know the
console is the PS5, I need to find
the manufacturer of its graphics
processing unit (GPU). </reasoning>
<conclusion> AMD </conclusion>
</step>
<step>
<reasoning> I have identified the
company as AMD. Now I need to find
its current CEO. </reasoning>
<search> current CEO of AMD </search>
<context> The current President
and CEO of AMD is Dr.  Lisa Su.
</context>
<conclusion> Dr.  Lisa Su
</conclusion>
</step>
<step>
<reasoning> I have the company (AMD)
and the CEO (Lisa Su).  Now I need
to find the stock price.  Today is
Saturday, July 19, 2025, so I need the
closing price from Friday, July 18,
2025.  </reasoning>
<search> AMD stock price closing July
18 2025 </search>
<context> AMD (Advanced Micro Devices,
Inc.)  closed at $175.40 on Friday,
July 18, 2025.  </context>
<conclusion> $175.40 </conclusion>
</step>
</think>
<answer>
Dr.  Lisa Su and $175.40.
</answer>
```

Figure 3: Comparison of reasoning trajectory formats for the same multi-hop question. Each logical step is highlighted in a consistent color across both formats to show the correspondence. The actual retrieved documents here are replaced by their summarization to improve readability.

# B    SYMBOLS

| Symbol | Description |
|---|---|
| $T$ | A complete reasoning trajectory for a given question. |
| $a$ | The final answer within a trajectory $T$. |
| $s_i$ | The i-th step in a reasoning trajectory. |
| $s^R$ | A search step, represented as a tuple $(r_i, q_i, c_i, o_i)$. |
| $s^{NR}$ | A non-search (internal reasoning) step, represented as a tuple $(r_i, o_i)$. |
| $r_i$ | The reasoning block of a step, containing planning and analysis. |
| $q_i$ | The search query generated in a search step. |
| $c_i$ | The retrieved context returned from a search query. |
| $o_i$ | The conclusion or summary of knowledge from a step. |
| $o'_i$ | The re-generated answer used for over-search detection. |
| $A(T)$ | A function indicating if the final answer $a$ is correct (1) or not (0). |
| $F(T)$ | A function indicating if the trajectory $T$ follows the required format (1) or not (0). |
| $N(T)$ | The total number of steps in a trajectory $T$. |
| $N_{\text{corr}}(T)$ | The number of optimal (correct) steps in trajectory $T$. |
| $\text{Over}(\cdot)$ | The detector function that identifies an over-search step. |
| $\text{Under}(\cdot)$ | The detector function that identifies an under-search step. |
| $\lambda_f$ | A hyperparameter representing the format weight. |
| $\lambda_p$ | A hyperparameter representing the process bonus coefficient. |
| $R(T)$ | The final hierarchical reward calculated for a trajectory $T$. |

Table 3: Description of symbols and notations used.

# C    ALGORITHMS

---
**Algorithm 1** Inference with Parsable Steps

---
**Require:** Question $q$, model $\pi$, retriever $Retrieve$, step budget $B$, top-$K$, preprocessed input (including instructions and user question) $y$

**Ensure:** Transcript with single `<think>` container and final `<answer>`

1: $y \leftarrow$ `<think><step><reasoning>`;   $i \leftarrow 1$
2: **while** $i \leq B$ **do**
3:     $\Delta \leftarrow$ GENERATE$(\pi, y)$ **until** boundary $\in \{$`</search>`,`</conclusion>`,`</answer>`$\}$
4:     $y \leftarrow y \,\|\, \Delta$
5:     **if** boundary $=$ `</answer>` **then**
6:         **break**                                                          ▷ final answer already produced
7:     **else if** boundary $=$ `</search>` **then**
8:         $s \leftarrow$ LASTBETWEEN$(y,$ `<search>`, `</search>`$)$
9:         $D \leftarrow Retrieve(s, K)$;   $c \leftarrow$ FORMAT$(D)$
10:         $y \leftarrow y \,\|\,$ `<context>` $\|\, c \,\|\,$ `</context><conclusion>`
11:         **continue**                                                      ▷ remain in the same step
12:     **else if** boundary $=$ `</conclusion>` **then**
13:         $y \leftarrow y \,\|\,$ `</step>`
14:         $i \leftarrow i + 1$
15:         **if** $i \leq B$ **then**
16:             $y \leftarrow y \,\|\,$ `<step><reasoning>`
17:         **end if**
18:     **end if**
19: **end while**
20: $y \leftarrow y \,\|\,$ `</think>`
21: **if not** CONTAINS$(y,$ `</answer>`$)$ **then**
22:     $y \leftarrow y \,\|\,$ `<answer>` $\|$ GENERATE$(\pi, y)$ **until** `</answer>`
23: **end if**
24: **return** $y$

---

---

**Algorithm 2** Format Checker $F(T)$

---

**Require:** Model output $y$ (string)
**Ensure:** $F(T) \in \{0, 1\}$ and (optionally) $N(T)$, the number of steps; return $(0, -1)$ on failure
1: $y \leftarrow \text{NORMALIZE}(y)$                ▷ canonicalize newlines; keep text intact
2: **if** $\text{COUNT}(y, \texttt{<think>}) \neq 1$ **or** $\text{COUNT}(y, \texttt{</think>}) \neq 1$ **then return** $(0, -1)$
3: $(i_{\text{open}}, i_{\text{close}}) \leftarrow \text{SPANBETWEEN}(y, \texttt{<think>}, \texttt{</think>})$      ▷ indices of content
4: **if** $\text{TRIM}(y[0 : i_{\text{open}}]) \neq \epsilon$ **then return** $(0, -1)$      ▷ no stray text before $\texttt{<think>}$
5: $t \leftarrow \text{BETWEEN}(y, \texttt{<think>}, \texttt{</think>})$          ▷ the content inside $\texttt{<think>}$
6: $post \leftarrow y[i_{\text{close}} + |\texttt{</think>}| :]$
7: **if not** $\text{STARTSWITHAFTERWS}(post, \texttt{<answer>})$ **then return** $(0, -1)$
8: **if** $\text{COUNT}(post, \texttt{<answer>}) \neq 1$ **or** $\text{COUNT}(post, \texttt{</answer>}) \neq 1$ **then return** $(0, -1)$
9: $ans \leftarrow \text{BETWEEN}(post, \texttt{<answer>}, \texttt{</answer>})$
10: **if** $\text{TRIM}(ans) = \epsilon$ **then return** $(0, -1)$
11: **if** $\text{TRIM}(post$ after $\texttt{</answer>}) \neq \epsilon$ **then return** $(0, -1)$      ▷ no trailing junk

12:                ▷ Inside $\texttt{<think>}$, allow only a sequence of well-formed $\texttt{<step>}$ blocks
13: $p \leftarrow 0; \quad N \leftarrow 0$
14: **while** True **do**
15:      $j \leftarrow \text{FINDFROM}(t, \texttt{<step>}, p)$
16:      **if** $j = -1$ **then**
17:          **if** $\text{TRIM}(t[p :]) \neq \epsilon$ **then return** $(0, -1)$      ▷ no stray text between steps
18:          **break**
19:      **end if**
20:      **if** $\text{TRIM}(t[p : j]) \neq \epsilon$ **then return** $(0, -1)$      ▷ no text before next $\texttt{<step>}$
21:      $k \leftarrow \text{FINDFROM}(t, \texttt{</step>}, j)$
22:      **if** $k = -1$ **then return** $(0, -1)$
23:      $s \leftarrow t[j + |\texttt{<step>}| : k]$          ▷ step body
24:      **if not** $\text{VALIDATESTEP}(s)$ **then return** $(0, -1)$
25:      $p \leftarrow k + |\texttt{</step>}|; \quad N \leftarrow N + 1$
26: **end while**
27: **if** $N < 1$ **then return** $(0, -1)$      ▷ must have at least one step
28: **return** $(1, N)$

---

**Algorithm 3** VALIDATESTEP - accepts exactly two step schemas (with or without search)

---

**Require:** Step body $s$ (string)
**Ensure:** **True** iff $s$ matches one of the allowed formats below and tags are unique, ordered, and properly nested
1: $s \leftarrow \text{STRIP}(s)$
2: **if** $\text{COUNT}(s, \texttt{<reasoning>}) \neq 1$ **or** $\text{COUNT}(s, \texttt{</reasoning>}) \neq 1$ **then return** False
3: $(r_o, r_c) \leftarrow \text{SPANBETWEEN}(s, \texttt{<reasoning>}, \texttt{</reasoning>})$
4: **if** $\text{TRIM}(s[0 : r_o]) \neq \epsilon$ **then return** False      ▷ $\texttt{<reasoning>}$ must be first
5: **if** $\text{COUNT}(s, \texttt{<conclusion>}) \neq 1$ **or** $\text{COUNT}(s, \texttt{</conclusion>}) \neq 1$ **then return** False
6: $(c_o, c_c) \leftarrow \text{SPANBETWEEN}(s, \texttt{<conclusion>}, \texttt{</conclusion>})$
7: **if** $c_o \leq r_c$ **then return** False      ▷ $\texttt{<conclusion>}$ must come after $\texttt{</reasoning>}$
8: **if** $\text{TRIM}(s[c_c + |\texttt{</conclusion>}| :]) \neq \epsilon$ **then return** False    ▷ step must end at $\texttt{</conclusion>}$

9:      ▷ Case A: step *with* search: $\texttt{<reasoning>} \rightarrow \texttt{<search>} \rightarrow \texttt{<context>} \rightarrow \texttt{<conclusion>}$
10: **if** $\text{CONTAINS}(s, \texttt{<search>})$ **or** $\text{CONTAINS}(s, \texttt{<context>})$ **then**
11:      **if** $\text{COUNT}(s, \texttt{<search>}) \neq 1$ **or** $\text{COUNT}(s, \texttt{</search>}) \neq 1$ **then return** False
12:      **if** $\text{COUNT}(s, \texttt{<context>}) \neq 1$ **or** $\text{COUNT}(s, \texttt{</context>}) \neq 1$ **then return** False
13:      $(q_o, q_c) \leftarrow \text{SPANBETWEEN}(s, \texttt{<search>}, \texttt{</search>})$
14:      $(x_o, x_c) \leftarrow \text{SPANBETWEEN}(s, \texttt{<context>}, \texttt{</context>})$
15:      **if** $r_c \geq q_o$ **or** $q_c \geq x_o$ **or** $x_c \geq c_o$ **then return** False      ▷ enforce strict order
16:      **if** $\text{TRIM}(s[r_c : q_o]) \neq \epsilon$ **or** $\text{TRIM}(s[q_c : x_o]) \neq \epsilon$ **or** $\text{TRIM}(s[x_c : c_o]) \neq \epsilon$ **then return** False
17:      **return** True
18: **end if**

19:      ▷ Case B: step *without* search: $\texttt{<reasoning>} \rightarrow \texttt{<conclusion>}$, and *no* $\texttt{<search>}/\texttt{<context>}$
20: **if** $\text{CONTAINS}(s, \texttt{<search>})$ **or** $\text{CONTAINS}(s, \texttt{<context>})$ **then return** False
21: **if** $\text{TRIM}(s[r_c : c_o]) \neq \epsilon$ **then return** False
22: **return** True

---

# D  PROMPTS

---

**System Prompt for Parsable Output Format**

Answer user questions by thinking step-by-step. Your entire reasoning process must be encapsulated within a single `<think></think>` block, which contains one or more `<step></step>` blocks. Each step must begin with your analysis in `<reasoning>`. If you identify a knowledge gap, you may use `<search>query</search>` to query a search engine; search results will then be provided in a `<context>` tag. Every step must end with a `<conclusion>` summarizing what you learned in that step. After your thinking process is complete, provide the final, conclusive answer inside an `<answer>` tag placed immediately after the closing `</think>` tag. You can use as many steps as you need. Ensure all XML tags are properly formed and nested.

**\*\*## Output Format Specification\*\***

Your output must follow this overall structure. The `<think>` block contains all the steps, and the `<answer>` block follows it.

```
<think>
<step>
...
</step>
<step>
...
</step>
</think>
<answer>Your final, conclusive answer to the user's
question.</answer>
```

---

**\*\*## Step Formats (to be used inside `<think>`)\*\***

**Format 1: Step with a Search**

```
<step>
<reasoning>Your detailed analysis...</reasoning>
<search>The precise search query...</search>
<context>[Provided by system]</context>
<conclusion>Your conclusion for this step.</conclusion>
</step>
```

**Format 2: Step without a Search (Internal Reasoning)**

```
<step>
<reasoning>Your detailed analysis...</reasoning>
<conclusion>Your conclusion for this step.</conclusion>
</step>
```

Figure 4: Input prompt for generating HiPRAG's parsable output format with the new XML tagging system.

**Prompt for Over-search Detection**

You are an expert in Natural Language Understanding and Semantic Analysis. Your goal is to determine if these two statements are semantically equivalent-that is, if they mean the same thing and convey the same core information. Provide your answers with a single boolean value "True" or "False" in the tag `<answer></answer>` (e.g. `<answer>True</answer>` or `<answer>False</answer>`).

Figure 5: Prompt for Over-search Detection

**Prompt for Under-search Detection**

You are an expert Fact-Checker and Logic Verifier. Your task is to evaluate a single, isolated reasoning step from an AI agent.

This step was generated without using a search tool. Your goal is to determine if the agent made a mistake by not searching, based only on the information within this single step and your own general knowledge.

Analyze the provided step by asking two questions:

1. **Factual Accuracy:** Is the statement in the `<reasoning></reasoning>` and `<conclusion></conclusion>` factually correct?

2. **Internal Logic:** Does the `<conclusion></conclusion>` logically follow from the `<reasoning></reasoning>` provided within this same step?

If both questions are answered correctly, provide your answers with a single boolean value "True" or "False" in the tag `<answer></answer>` (e.g. `<answer>True</answer>` or `<answer>False</answer>`).

Figure 6: Prompt for Under-search Detection

# E   DETAILED REPORT FOR CEM, OSR, AND USR

| Base Model | RL Algo. + Method | General QA | | | Multi-Hop QA | | | | Avg. |
|---|---|---|---|---|---|---|---|---|---|
| | | NQ | TriviaQA | PopQA | HotpotQA | 2Wiki | MuSiQue | Bamboogle | |
| Llama-3.2-3B-Instruct | PPO + baseline | 65.2 | 74.5 | 55.1 | 47.0 | 52.3 | 18.7 | 36.0 | 56.4 |
| Llama-3.2-3B-Instruct | PPO + HiPRAG | **71.6** | **77.2** | 61.0 | 57.7 | 67.9 | 25.7 | 43.2 | 64.8 |
| Qwen2.5-3B-Instruct | GRPO + baseline | 59.6 | 69.1 | 57.3 | 52.4 | 61.4 | 20.6 | 24.8 | 58.5 |
| Qwen2.5-3B-Instruct | GRPO + HiPRAG | 68.5 | 74.2 | 60.6 | 59.2 | 69.1 | 27.9 | 38.4 | 64.4 |
| Qwen2.5-3B | PPO + baseline | 60.6 | 71.7 | 55.8 | 54.3 | 65.7 | 24.1 | 40.8 | 60.3 |
| Qwen2.5-3B | PPO + HiPRAG | 68.7 | 75.5 | **66.3** | 57.4 | 67.4 | 24.1 | 41.6 | 65.4 |
| Qwen2.5-3B-Instruct | PPO + baseline | 60.9 | 70.1 | 57.0 | 52.0 | 63.0 | 24.3 | 37.6 | 59.3 |
| Qwen2.5-3B-Instruct | PPO + HiPRAG | 65.6 | 73.9 | 62.1 | 55.6 | 69.6 | 26.0 | 32.8 | 64.1 |
| Qwen2.5-7B-Instruct | GRPO + baseline | 62.4 | 74.4 | 57.3 | 54.8 | 64.2 | 25.3 | 49.6 | 61.2 |
| Qwen2.5-7B-Instruct | GRPO + HiPRAG | 71.2 | 76.3 | 63.2 | **62.4** | **71.7** | 34.1 | **52.8** | **67.2** |
| Qwen2.5-7B-Instruct | PPO + baseline | 55.6 | 67.5 | 43.5 | 49.4 | 58.5 | 26.6 | 44.0 | 53.3 |
| Qwen2.5-7B-Instruct | PPO + HiPRAG | 66.2 | 75.7 | 58.4 | 59.9 | 66.2 | **34.3** | 52.0 | 64.5 |
| Qwen2.5-3B-Instruct | PPO + HiPRAG (over-search only) | 61.9 | 66.9 | 54.9 | 52.2 | 65.4 | 25.5 | 39.2 | 58.8 |
| Qwen2.5-3B-Instruct | PPO + HiPRAG (under-search only) | 63.7 | 74.1 | 60.6 | 55.9 | 67.9 | 28.4 | 40.8 | 63.3 |
| Qwen2.5-3B-Instruct | PPO + HiPRAG ($\lambda_p = 0.2$) | 61.9 | 71.2 | 56.8 | 53.7 | 62.2 | 25.4 | 31.2 | 59.6 |
| Qwen2.5-3B-Instruct | PPO + HiPRAG ($\lambda_p = 0.6$) | 66.6 | 74.4 | 60.5 | 55.5 | 64.4 | 25.6 | 38.4 | 62.5 |

Table 4: Cover Exact Match in percentage of HiPRAG models trained with different parameters on seven QA benchmarks. Best and second-best results for each dataset are marked in **bold** and underline. Baseline here refers to the reward with process bonus disabled ($\lambda_p = 0$).

| Base Model | RL Algo. + Method | General QA | | | Multi-Hop QA | | | | Avg. |
|---|---|---|---|---|---|---|---|---|---|
| | | NQ | TriviaQA | PopQA | HotpotQA | 2Wiki | MuSiQue | Bamboogle | |
| Llama-3.2-3B-Instruct | PPO + baseline | 12.5 | 15.4 | 5.0 | 4.8 | 3.7 | 2.7 | 8.7 | 7.3 |
| Llama-3.2-3B-Instruct | PPO + HiPRAG | 11.9 | 13.3 | 4.5 | 4.6 | 1.8 | 3.1 | 5.0 | 6.0 |
| Qwen2.5-3B-Instruct | GRPO + baseline | 8.4 | 17.0 | 5.6 | 7.2 | 4.3 | 5.0 | 10.3 | 8.4 |
| Qwen2.5-3B-Instruct | GRPO + HiPRAG | 4.4 | 9.8 | 2.2 | 3.0 | 2.9 | 1.4 | 3.9 | 4.1 |
| Qwen2.5-3B | PPO + baseline | 6.4 | 9.0 | 2.6 | 2.9 | 1.5 | 1.7 | 4.3 | 3.8 |
| Qwen2.5-3B | PPO + HiPRAG | 5.1 | 6.9 | 2.2 | 2.3 | 1.4 | **1.2** | **3.4** | 3.2 |
| Qwen2.5-3B-Instruct | PPO + baseline | 8.6 | 13.5 | 5.6 | 4.2 | 1.8 | 3.9 | 12.8 | 6.1 |
| Qwen2.5-3B-Instruct | PPO + HiPRAG | 6.0 | 11.0 | 3.9 | 4.5 | 2.5 | 2.8 | 11.5 | 4.9 |
| Qwen2.5-7B-Instruct | GRPO + baseline | 5.3 | 7.4 | 2.0 | 3.5 | 0.9 | 3.6 | 8.7 | 5.2 |
| Qwen2.5-7B-Instruct | GRPO + HiPRAG | **4.1** | **5.4** | **1.3** | **1.8** | **0.3** | 1.5 | 4.8 | **2.3** |
| Qwen2.5-7B-Instruct | PPO + baseline | 11.6 | 19.8 | 6.1 | 7.4 | 2.7 | 8.5 | 19.6 | 7.6 |
| Qwen2.5-7B-Instruct | PPO + HiPRAG | 10.4 | 14.5 | 4.6 | 5.6 | 2.1 | 5.9 | 13.4 | 6.2 |
| Qwen2.5-3B-Instruct | PPO + HiPRAG (over-search only) | 6.0 | 11.0 | 3.9 | 4.5 | 2.5 | 2.8 | 11.5 | 4.9 |
| Qwen2.5-3B-Instruct | PPO + HiPRAG (under-search only) | 8.2 | 15.6 | 5.9 | 5.3 | 2.7 | 3.1 | 6.4 | 6.6 |
| Qwen2.5-3B-Instruct | PPO + HiPRAG ($\lambda_p = 0.2$) | 7.4 | 11.4 | 3.7 | 4.4 | 2.3 | 3.3 | 12.1 | 5.5 |
| Qwen2.5-3B-Instruct | PPO + HiPRAG ($\lambda_p = 0.6$) | 9.3 | 13.1 | 4.1 | 3.3 | 1.3 | 2.2 | 6.3 | 5.2 |

Table 5: Over-search Rate in percentage (lower is better) on seven QA benchmarks. Best and second-best values for each dataset are marked in **bold** and underline. Baseline here refers to the reward with process bonus disabled ($\lambda_p = 0$).

| Base Model | RL Algo. + Method | General QA | | | Multi-Hop QA | | | | Avg. |
|---|---|---|---|---|---|---|---|---|---|
| | | NQ | TriviaQA | PopQA | HotpotQA | 2Wiki | MuSiQue | Bamboogle | |
| Llama-3.2-3B-Instruct | PPO + baseline | 67.1 | 75.0 | 66.7 | 52.6 | 59.3 | 50.0 | 20.0 | 57.6 |
| Llama-3.2-3B-Instruct | PPO + HiPRAG | 35.3 | 48.4 | 31.7 | 50.8 | 55.3 | 64.3 | 10.3 | 49.7 |
| Qwen2.5-3B-Instruct | GRPO + baseline | 61.9 | 63.9 | 59.6 | 46.1 | 49.1 | 61.9 | 22.3 | 52.1 |
| Qwen2.5-3B-Instruct | GRPO + HiPRAG | 52.9 | 34.9 | 35.2 | 29.2 | 25.0 | 45.5 | 21.2 | 33.2 |
| Qwen2.5-3B | PPO + baseline | 33.3 | 66.7 | 30.8 | 38.5 | 47.5 | 66.7 | **0.0** | 44.0 |
| Qwen2.5-3B | PPO + HiPRAG | 43.9 | 36.4 | 42.3 | 41.9 | 42.6 | 56.8 | 16.7 | 41.9 |
| Qwen2.5-3B-Instruct | PPO + baseline | 47.1 | 33.2 | 48.8 | 39.0 | 52.9 | 70.0 | 32.2 | 47.5 |
| Qwen2.5-3B-Instruct | PPO + HiPRAG | **11.1** | 44.4 | 61.9 | 25.0 | 32.0 | **10.1** | 8.7 | 38.1 |
| Qwen2.5-7B-Instruct | GRPO + baseline | 40.5 | 34.3 | 43.8 | 40.9 | 45.0 | 56.2 | 20.0 | 43.3 |
| Qwen2.5-7B-Instruct | GRPO + HiPRAG | 30.2 | 34.9 | 34.9 | 40.5 | 24.4 | 37.3 | 41.7 | 32.6 |
| Qwen2.5-7B-Instruct | PPO + baseline | 33.4 | **13.9** | 17.5 | 40.3 | 33.4 | 50.0 | 13.2 | 33.9 |
| Qwen2.5-7B-Instruct | PPO + HiPRAG | 57.1 | 44.9 | 25.5 | **20.0** | 34.6 | 57.1 | 1.6 | 29.0 |
| Qwen2.5-3B-Instruct | PPO + HiPRAG (over-search only) | 54.5 | 55.2 | 48.9 | 44.7 | 53.7 | 78.3 | 20.0 | 52.7 |
| Qwen2.5-3B-Instruct | PPO + HiPRAG (under-search only) | 14.0 | 20.4 | **13.6** | 25.6 | **13.2** | 30.8 | 16.9 | **16.9** |
| Qwen2.5-3B-Instruct | PPO + HiPRAG ($\lambda_p = 0.2$) | 24.0 | 39.5 | 55.8 | 41.3 | 45.5 | 80.1 | 30.1 | 44.5 |
| Qwen2.5-3B-Instruct | PPO + HiPRAG ($\lambda_p = 0.6$) | 27.2 | 34.1 | 33.6 | 60.6 | 51.2 | 53.3 | 1.6 | 39.0 |

Table 6: Under-search Rate in percentage (lower is better) on seven QA benchmarks. Best and second-best results for each dataset are marked in **bold** and underline. Baseline here refers to the reward with process bonus disabled ($\lambda_p = 0$).

## F    ADDITIONAL ANALYSIS ON EFFICACY

### F.1    FORMAT CORRECTNESS PERCENTAGE ANALYSIS

To verify the effectiveness of our prompting and reward strategy for enforcing a machine-parsable output, we analyzed the format correctness across all test samples generated by our final HiPRAG-trained models. Our analysis shows that 96.3% of all generated trajectories successfully adhered to the required format. This high percentage confirms that the models effectively learned to produce the structured output, which is a critical prerequisite for the successful application of our on-the-fly process reward mechanism.

### F.2    EFFICACY OF OVER-SEARCH & UNDER-SEARCH DETECTION

The reliability of our process-based reward hinges on the accuracy of the over-search and under-search detection methods. To validate these, we manually inspected the detection results for 200 randomly selected reasoning trajectories from our test set evaluations. The manual audit revealed a 98.3% accuracy rate for over-search detection and a 95.6% accuracy rate for under-search detection. These high accuracy figures confirm that our on-the-fly LLM-based judges provide a reliable and effective signal for identifying suboptimal search behaviors during RL training.

### F.3    EFFICACY OF CEM METRIC

To ensure our primary evaluation metric, Cover Exact Match (CEM), accurately reflects model performance, we manually inspected its judgments on 100 randomly sampled question-answer pairs from our test results. Our review found that the CEM metric's assessment of correctness aligned with human judgment in 98% of cases. This confirms that CEM is a robust metric for this task, appropriately handling valid answers embedded within the longer, more explanatory responses typical of modern LLMs, thus avoiding unfairly penalizing models for verbosity.

## G    ADDITIONAL STUDY ON ROBUSTNESS TO LLM JUDGE CHOICE

To study sensitivity to external LLM judges used for on-the-fly detection, we conducted an experiment replacing our standard proprietary LLMs (GPT-4.1 mini (OpenAI, 2025a) and GPT-5 mini (OpenAI, 2025b)) with open-source models of varying capabilities, specifically the Qwen3-30B-A3B-Instruct/Thinking-2507 and Qwen3-4B-Instruct/Thinking-2507 (Instruct for over-search, Thinking for under-search) (Yang et al., 2025). We use greedy decoding with fixed random seed for all 4 judges to ensure stability and reproducibility. For other parameters, we maintain the exact experimental setup as our main Qwen2.5-3B-Instruct + PPO experiment.

| LLM Judges | Avg. CEM | Avg. OSR | Avg. USR |
|---|---|---|---|
| Standard (GPT) | 64.1 | 4.9 | 38.1 |
| Strong Open (30B) | 64.0 | 4.8 | 39.2 |
| Weaker Open (4B) | 63.4 | 4.9 | 42.4 |

Table 7: Summary of Performance (Avg. across 7 Benchmarks).

### G.1    ANALYSIS

**1. Sensitivity of Over-Search Detection (OSR):** The OSR remained nearly constant across all judge configurations. This suggests that the task of detecting semantic equivalence is robust and can be handled effectively even by smaller, non-reasoning "Instruct" models. The definition of redundancy does not require complex reasoning chains, making it less sensitive to model size.

**2. Sensitivity of Under-Search Detection (USR):** The USR showed higher sensitivity. The weaker Qwen3-4B judges resulted in a USR increase from 38.1% to 42.4%. We hypothesize that weaker judges are less rigorous in factual/logical verification, occasionally failing to flag "hallucinated"

reasoning or premature conclusions. This allows the agent to terminate the search process earlier than optimal, leading to the slight degradation in the final accuracy (CEM).

**3. Performance Degradation:** Despite the increase in USR with weaker judges, the overall degradation is not catastrophic. This confirms that the hierarchical process reward framework itself provides a stable training signal even with sub-optimal judges.

## G.2 DETAILED RESULTS BY DATASET

Below we provide the detailed breakdown of performance across all seven QA benchmarks used in the paper.

| LLM Judges | NQ | TriviaQA | PopQA | HotpotQA | 2Wiki | MuSiQue | Bamboogle | Avg. |
|---|---|---|---|---|---|---|---|---|
| Standard (GPT) | 65.6 | 73.9 | 62.1 | 55.6 | 69.6 | 26.0 | 32.8 | 64.1 |
| Strong Open (30B) | 65.3 | 73.6 | 62.9 | 55.2 | 69.2 | 26.4 | 32.4 | 64.0 |
| Weaker Open (4B) | 64.7 | 73.7 | 60.8 | 56.5 | 69.5 | 24.4 | 31.5 | 63.4 |

Table 8: Cover Exact Match (CEM) by Dataset.

| LLM Judges | NQ | TriviaQA | PopQA | HotpotQA | 2Wiki | MuSiQue | Bamboogle | Avg. |
|---|---|---|---|---|---|---|---|---|
| Standard (GPT) | 6.0 | 11.0 | 3.9 | 4.5 | 2.5 | 2.8 | 11.5 | 4.9 |
| Strong Open (30B) | 5.9 | 10.8 | 3.8 | 4.8 | 2.4 | 1.7 | 11.9 | 4.8 |
| Weaker Open (4B) | 5.7 | 9.5 | 4.2 | 4.6 | 3.9 | 2.6 | 11.3 | 4.9 |

Table 9: Over-Search Rate (OSR) by Dataset.

| LLM Judges | NQ | TriviaQA | PopQA | HotpotQA | 2Wiki | MuSiQue | Bamboogle | Avg. |
|---|---|---|---|---|---|---|---|---|
| Standard (GPT) | 11.1 | 44.4 | 61.9 | 25.0 | 32.0 | 10.1 | 8.7 | 38.1 |
| Strong Open (30B) | 11.9 | 48.1 | 63.0 | 22.1 | 40.4 | 10.9 | 5.3 | 39.2 |
| Weaker Open (4B) | 10.7 | 54.0 | 60.3 | 27.2 | 45.5 | 18.5 | 10.3 | 42.4 |

Table 10: Under-Search Rate (USR) by Dataset.

## G.3 ANALYSIS OF JUDGMENT ERRORS AND ROBUSTNESS TO JUDGE SELECTION

To address the reviewer's concern about the judgment errors, we performed a manual evaluation of judgment accuracy and analyzed the sensitivity of our method to these errors.

**Judge Accuracy:**  Following the protocol in Appendix F, we sampled 200 trajectories from the test set and established human-labeled ground truth for step-level Over-Search and Under-Search decisions, for all the steps in those trajectories.

As shown in Table 11, all models maintain high agreement with human judgment.

**Impact of Judgment Errors:**  We analyzed the correlation between judge accuracy and downstream agent performance to quantify the impact of mislabeling:

- **Under-search:** The Under-search judge is the primary driver of performance variance. The main risk is a False Negative (failing to flag an under-search). This effectively disrupts the reasoning trajectory, causing the agent to stop searching prematurely, or directly leads to incorrect final results.

- **Over-search:** Over-search mislabels typically result in a step being flagged as redundant (or not) without altering the information available to the agent. Consequently, these errors rarely change the final answer path, resulting in minimal impact on overall accuracy.

| External LLM Judges | Over-search Accuracy | Under-search Accuracy |
|---|---|---|
| Standard (GPT) | 98.3% | 95.6% |
| Strong Open (30B) | 97.5% | 94.5% |
| Weaker Open (4B) | 96.5% | 92.5% |

Table 11: Judge Accuracy against Human Ground Truth.

## H  ADDITIONAL STUDY ON AVERAGE NUMBER OF SEARCHES

In addition to OSR and USR, we report the average number of search steps per question (Avg. #Searches) as a raw efficiency metric:

$$\text{Avg. \#Searches} = \frac{\sum_{T \in \mathcal{D}_{\text{test}}} \left| \{ s^R \in T \} \right|}{|\mathcal{D}_{\text{test}}|}. \tag{3}$$

For a fair comparison, this table uses the same Qwen2.5-3B-Instruct + PPO for all methods.

| Method | 2Wiki | Bamboogle | HotpotQA | MuSiQue | NQ | PopQA | TriviaQA | Avg. |
|---|---|---|---|---|---|---|---|---|
| Search-R1 | 3.38 | 2.79 | 2.78 | 3.39 | 1.55 | 1.75 | 1.53 | 2.45 |
| $\beta$-GRPO | 2.85 | 2.15 | 2.45 | 2.50 | 1.28 | 1.50 | 0.90 | 2.15 |
| HiPRAG (Ours) | 2.48 | 2.02 | 2.14 | 2.55 | 1.01 | 1.02 | 1.01 | 1.75 |

Table 12: Average number of searches per question (lower is better).

HiPRAG's step-wise process reward achieves good efficiency by evaluating the necessity of each step on-the-fly, consistently using the fewest searches while maintaining higher accuracy. We also emphasize that the Over-search Rate (OSR) and Under-search Rate (USR) are critical efficiency metrics. Naive penalties on trajectory length or search frequency often cause models to avoid searching even when necessary (increasing under-search), which leads to performance degradation.

## I  COMPUTATIONAL TIME ANALYSIS DURING RL TRAINING

We profile the RL training pipeline to quantify the overhead introduced by HiPRAG's on-the-fly reward assignment. The profiling is conducted on Qwen2.5-3B-Instruct + PPO using 8 NVIDIA A100 80GB GPUs (veRL with vLLM inference and FSDP training).

| Component | Time (s) | Percent |
|---|---|---|
| Rollout generation | 206.24 | 63.55% |
| PPO update | 46.19 | 14.23% |
| Reference & critic inference | 30.09 | 9.27% |
| HiPRAG reward assignment (re-generation & judge calls) | 42.03 | 12.95% |
| **Total** | 324.55 | 100% |

Table 13: Runtime breakdown of one RL training step (wall-clock).

The specific overhead introduced by HiPRAG (the batched re-generation and external API calls) accounts for only 12.95% of the total training time. This is a small increase, especially when weighed against the significant gains in sample efficiency and final model performance.

To effectively reduce overhead in the extra process of reward computation, we utilize the following methods during training.

- **Batched Generation:** The "re-generation" check for over-search is performed via batch generation. We collect the queries from the rollout, aggregate them into several batches, and process the queries in each batch in parallel.

- **Targeted Scope:** Re-generation is only triggered for search steps, not every step in the trajectory.

- **Generation Length:** The re-generation prompts the model for a direct answer or relevant information to the search query, which is significantly shorter than the full reasoning chain generated during rollout even without actively limiting the generation length.

- **Asynchronous API Calls:** As noted in Section 3.2, the external detection is executed separately. The batch processing of API calls to the external verifier occurs in parallel with the local model's batched re-generation, therefore effectively reducing the latency. Specifically, for over-search detection, the verification calls to external LLM judges start once the first batch has finished and continue concurrently in the later batches (for example, for re-generation batch $t$, execute batch $t-1$ verifier API calls); for under-search detection, these verification calls to external LLM judges can be executed concurrently with the re-generation of over-search detection as they are independent processes.

- **Cost Reduction for LLM judges:** We utilize cost-effective models for verification. On top of that, we use a non-reasoning model for over-search detection for further cost saving, because the nature of over-search detection is checking the equivalence of the conclusion of the search step and the re-generation result, which does not require complex reasoning abilities. For under-search detection, due to the need of detecting logical error in the non-search step, a reasoning model is required.

## J ADDITIONAL STUDY FOR FORMAT REWARD WEIGHT $\lambda_f$

To address the impact of the format reward, we conducted a study varying $\lambda_f \in \{0.2, 0.4, 0.6\}$ using the Qwen2.5-3B-Instruct + PPO setup, keeping the process bonus fixed at $\lambda_p = 0.4$.

The results clearly demonstrate that our chosen value of $\lambda_f = 0.2$ provides the optimal trade-off between answer accuracy (Avg. CEM) and search efficiency (Avg. OSR/USR). As shown in Table 14, the $\lambda_f = 0.2$ setting achieves the highest accuracy and the best efficiency scores, while larger values over-emphasize the instrumental goal of format correctness and degrade performance.

| $\lambda_f$ | Avg. CEM ↑ | Avg. OSR ↓ | Avg. USR ↓ | Observation |
|---|---|---|---|---|
| 0.2 | 64.1 | 4.9 | 38.1 | Best trade-off of accuracy and efficiency. |
| 0.4 | 63.2 | 5.2 | 37.9 | Degraded accuracy and efficiency. |
| 0.6 | 61.2 | 5.4 | 41.0 | Over-emphasis on format harms performance. |

Table 14: Changing format weight ($\lambda_f$) with $\lambda_p = 0.4$ fixed. The $\lambda_f = 0.2$ setting used in our paper achieves the highest accuracy and best efficiency scores.

### J.1 SENSITIVITY ANALYSIS AND DISCUSSION

This trend is expected, as it aligns with our core intuition that answer correctness is significantly more important than format correctness. We justify this principle as follows:

- **Terminal Goal vs. Instrumental Goal:** Format correctness ($F(T)$) is an instrumental goal. Its primary purpose is to make the agent's reasoning trajectory parsable, which in turn enables our hierarchical process reward to be calculated. In contrast, answer correctness ($A(T)$) is the terminal goal of the entire agentic RAG task. The user's objective is to get the right answer; a well-formatted but incorrect trajectory is ultimately a failure. To prioritize answer correctness, the format weight $\lambda_f$ must be small.

- **Preventing "Reward Hacking":** If $\lambda_f$ were set too high, the agent could "hack" the reward function by learning to produce perfectly formatted, minimal-step trajectories (e.g., a single non-search step) that are factually incorrect. This would maximize the $F(T)$ component of the reward while ignoring $A(T)$ and the process bonus. A low $\lambda_f$ (like 0.2) ensures the reward landscape is always dominated by the incentive to get the answer right. This principle is directly reflected in our hierarchical reward function.

| $\lambda_f$ | NQ | TriviaQA | PopQA | HotpotQA | 2Wiki | MuSiQue | Bamboogle | Avg. |
|---|---|---|---|---|---|---|---|---|
| | | | CEM ($\uparrow$) by dataset (%) under $\lambda_f$ ablation | | | | | |
| 0.2 | 65.6 | 73.9 | 62.1 | 55.6 | 69.6 | 26.0 | 32.8 | 64.1 |
| 0.4 | 64.7 | 72.7 | 61.3 | 55.8 | 68.4 | 26.5 | 33.5 | 63.2 |
| 0.6 | 63.5 | 71.5 | 60.0 | 53.9 | 67.2 | 23.8 | 30.8 | 61.2 |
| | | | OSR ($\downarrow$) by dataset (%) under $\lambda_f$ ablation | | | | | |
| 0.2 | 6.0 | 11.0 | 3.9 | 4.5 | 2.5 | 2.8 | 11.5 | 4.9 |
| 0.4 | 6.3 | 11.4 | 4.0 | 3.7 | 3.0 | 3.9 | 11.2 | 5.2 |
| 0.6 | 6.1 | 11.0 | 4.4 | 5.1 | 2.9 | 3.2 | 13.0 | 5.4 |
| | | | USR ($\downarrow$) by dataset (%) under $\lambda_f$ ablation | | | | | |
| 0.2 | 11.1 | 44.4 | 61.9 | 25.0 | 32.0 | 10.1 | 8.7 | 38.1 |
| 0.4 | 12.0 | 43.2 | 60.8 | 22.1 | 35.1 | 9.6 | 9.1 | 37.9 |
| 0.6 | 13.4 | 46.8 | 64.3 | 27.6 | 34.8 | 11.3 | 9.7 | 41.0 |

Table 15: Detailed performance breakdown for individual datasets under $\lambda_f$ ablation.

## K    CASE STUDY

To illustrate the practical benefits of our HiPRAG framework, we examine a specific case where the baseline model fails due to inefficient reasoning, while our trained agent succeeds. The models here are trained based on Qwen2.5-3B-Instruct. The question posed is: "What is the place of birth of the performer of song Slow Down (Lacy J. Dalton Song)?". The baseline model overlooks the crucial parenthetical information, "(Lacy J. Dalton Song)," and initiates a broad search for the song "Slow Down." This leads to an unnecessary five-step process where it identifies three different artists who have a song by that title and then searches for the birthplace of each one individually. Furthermore, through over-search detection, the answer of the final search step's query about the birthplace of Selena Gomez is answered correctly and equivalent to the original answer "Grand Prairie, Texas" in reasoning trajectory. The final answer of the baseline is incorrect due to the interruption of unnecessary searches. This is a classic example of over-searching, where the agent performs redundant and irrelevant lookups, ultimately failing to provide a single, correct answer. In contrast, the HiPRAG-trained agent correctly parses the entire question in its first, non-search step, identifying Lacy J. Dalton as the specified performer. It then executes a single, targeted search for her place of birth. This two-step, optimal reasoning path-one internal reasoning step followed by one necessary search-avoids the inefficiencies of the baseline, leading directly to the correct answer. This case clearly demonstrates how HiPRAG's process-oriented rewards cultivate a more nuanced and efficient reasoning strategy, improving both accuracy and search economy.

---

**Baseline Model (Inefficient Reasoning)**

**Question:** What is the place of birth of the performer of song Slow Down (Lacy J. Dalton Song)?
**Golden answers:** ['Bloomsburg, Pennsylvania', 'The Only Town in Pennsylvania', 'Bloomsburg']
**\<think\>**
**\<step\>**
**\<reasoning\>**I don't have direct information about the performer of the song 'Slow Down' and their place of birth.**\</reasoning\>**
**\<search\>**What is the performer of the song 'Slow Down'?**\</search\>**
**\<context\>**... [Returns Doc 1: Bobby Valentino, Doc 2: Douwe Bob, Doc 3: Selena Gomez] ...**\</context\>**
**\<conclusion\>**Based on the search results, 'Slow Down' is a song by three different performers: Bobby Valentino, Douwe Bob, and Selena Gomez. I don't have information about the place of birth of the performer of this song.**\</conclusion\>**
**\</step\>**
**\<step\>**
**\<reasoning\>**Since I don't have information about the performer's place of birth, I will first need to determine who the performer of the song 'Slow Down' is.**\</reasoning\>**
**\<search\>**Who is the performer of the song 'Slow Down'?**\</search\>**
**\<context\>**... [Returns Doc 1: Bobby Valentino, Doc 2: Douwe Bob, Doc 3: Selena Gomez] ...**\</context\>**
**\<conclusion\>**Based on the search results, the performer of the song 'Slow Down' can be either Bobby Valentino, Douwe Bob, or Selena Gomez.**\</conclusion\>**
**\</step\>**
**\<step\>**
**\<reasoning\>**Now that I know the performers, I will need to find out the place of birth of each of them...**\</reasoning\>**
**\<search\>**What is the place of birth of Bobby Valentino?**\</search\>**
**\<context\>**... [Returns birthplace: Chatham, Kent] ...**\</context\>**
**\<conclusion\>**Based on the search results, Bobby Valentino was born in Chatham, Kent.**\</conclusion\>**
**\</step\>**
**\<step\>**
**\<reasoning\>**Now that I have the place of birth of Bobby Valentino, I will need to find out the place of birth of Douwe Bob.**\</reasoning\>**
**\<search\>**What is the place of birth of Douwe Bob?**\</search\>**
**\<context\>**[Returns birthplace: Amsterdam, Netherlands]**\</context\>**
**\<conclusion\>**Based on the search results, Douwe Bob was born in Amsterdam, Netherlands.**\</conclusion\>**
**\</step\>**
**\<step\>**
**\<reasoning\>**Finally, I will need to find out the place of birth of Selena Gomez.**\</reasoning\>**
**\<search\>**What is the place of birth of Selena Gomez?**\</search\>**
**\<context\>**[Returns birthplace: Grand Prairie, Texas]**\</context\>**
**\<conclusion\>**Based on the search results, Selena Gomez was born in Grand Prairie, Texas.**\</conclusion\>**
**\</step\>**
**\</think\>**
**\<answer\>**Based on the information gathered, the performer of the song 'Slow Down' can be either Bobby Valentino, Douwe Bob, or Selena Gomez. Bobby Valentino was born in Chatham, Kent. Douwe Bob was born in Amsterdam, Netherlands. Selena Gomez was born in Grand Prairie, Texas.**\</answer\>**

Figure 7: Case study: Baseline reasoning trajectory. The model has five unnecessary search steps (**over-searching**) and an incorrect, ambiguous final answer.

---

**HiPRAG-trained Model (Efficient Reasoning)**

**Question:** What is the place of birth of the performer of song Slow Down (Lacy J. Dalton Song)?
**Golden answers:** ['Bloomsburg, Pennsylvania', 'The Only Town in Pennsylvania', 'Bloomsburg']
`<think>`
`<step>`
`<reasoning>`What do I need to know to answer this question? First, I need to find this performer of Slow Down. Then I need to find out the place of birth of the performer.`</reasoning>`
`<conclusion>`According to the question, the performer of the song Slow Down (Lacy J. Dalton Song) is Lacy J. Dalton...`</conclusion>`
`</step>`

`<step>`
`<reasoning>`Now that I have the answer to the first part, I need to find the place of birth of the performer. Where can I find this information? I will need to search for it.`</reasoning>`
`<search>`Place of birth of Lacy J. Dalton`</search>`
`<context>`Doc 1(Title: "Lacy J. Dalton") Lacy J. Dalton (born Jill Lynne Byrem; October 13, 1946, Bloomsburg, Pennsylvania) is an American country singer...`</context>`
`<conclusion>`According to the context, the place of birth of Lacy J. Dalton is Bloomsburg, Pennsylvania.`</conclusion>`
`</step>`
`</think>`

`<answer>`According to the information found, the place of birth of the performer of the song Slow Down (Lacy J. Dalton Song) is Bloomsburg, Pennsylvania. Lacy J. Dalton was born on October 13, 1946, in Bloomsburg, Pennsylvania.`</answer>`

Figure 8: Case study: HiPRAG-trained reasoning trajectory. The model correctly identifies the key entity in a non-search step, then executes a single, targeted search to find the correct answer efficiently.

## L    USE OF LARGE LANGUAGE MODELS

We acknowledge the use of the GPT-5 language model provided by OpenAI in the final stages of manuscript preparation. This tool was employed exclusively for identifying and correcting typographical and grammatical errors, ensuring clarity and precision in the written presentation. Its use was strictly limited to linguistic refinement and did not impact the study's conceptual framework, research methodology, data analysis, or conclusions. All intellectual contributions and substantive content remain those of the authors. The usage of all other LLMs (GPT-4.1 mini, GPT-5 mini, Qwen2.5, and Llama-3.2) mentioned in this work is part of the experiments themselves.

