# OpenReview forum: "HiPRAG: Hierarchical Process Rewards for Efficient Agentic Retrieval Augmented Generation"
_ICLR.cc/2026/Conference — ICLR 2026 Poster_

### Official Review · Reviewer_DZXL · 2025-10-29

**Soundness:** 3
**Presentation:** 4
**Contribution:** 4
**Rating:** 4
**Confidence:** 2

**Summary:**

The main contribution of this work is HiPRAG, a novel Reinforcement Learning training methodology that incorporates a fine-grained, knowledge-aware "Hierarchical Process Reward". This approach is achieved by first decomposing the agent's reasoning trajectory into discrete, parsable <step> blocks to distinguish between search and non-search steps ; it then evaluates the necessity of each search decision "on-the-fly" during training, using external LLM judges to identify "over-search" and "under-search" behaviors ; finally, it applies a hierarchical reward function that provides an additional "gated process bonus" based on the proportion of optimal steps, which is added only when both the final answer and the format are correct. Experiments show HiPRAG achieves 67.2% average accuracy (7B), outperforming strong baselines , while dramatically improving efficiency by reducing the over-search rate from over 27% to just 2.3%.

**Strengths:**

1.The quality of the experimental methodology is high. The paper's conclusions are based on extensive experiments across seven QA benchmarks, multiple model families (Qwen, Llama), and multiple RL algorithms (PPO, GRPO). The ablation studies in Section 5.3 are exhaustive and provide strong support for HiPRAG's key design choices.

2.A key finding is that the HiPRAG model trained based on Qwen2.5-3B-Instruct + GRPO not only surpasses strong external 7B baselines like R1-Searcher++, but it also outperforms the 7B counterpart trained with the authors' baseline reward . This indicates that the authors' training methodology is a more effective path to performance gains than simply scaling the model size with conventional rewards, which has significant implications for resource-constrained research and deployment.

3.The paper's presentation is very clear and well-organized, making a complex methodology easy to understand. The figures in the paper are of high quality and greatly enhance understanding.

**Weaknesses:**

The Over-search Detection method  requires the policy model to perform a second, complete generation. This "re-generation step" effectively doubles the inference cost for each search step during the RL rollout phase. Although the authors mention that "the re-generation step can be executed separately through batch generation", this still represents significant training overhead. A discussion on the impact of this training method on the total training time would be valuable.

**Questions:**

1.Why did the authors not provide an ablation study for λf ? How was the value of 0.2 determined? How sensitive are the model's final performance and convergence stability to the choice of λf ?

2.The knowledge source used for the experiments is the 2018 Wikipedia dump , while the judge model for 'under-search' detection is gpt-5-mini, which possesses 2025-era knowledge. If the policy model provides a non-search answer that was factually correct based on the 2018 knowledge base,but is outdated according to the 2025 judge's knowledge, will the judge incorrectly flag this step as an 'under-search'? Could this "temporal misalignment" between the retrieval knowledge source and the judge's internal knowledge base introduce significant noisy rewards, thereby contaminating the training process?

---

> ### Author Response · Authors · 2025-11-20
> **Response to Reviewer DZXL - Post (1)**
>
> ### Response to Weakness (Training Overhead)
> See common post [“Computational Time Analysis During RL Training”](https://openreview.net/forum?id=Gt4v9WBPzm&noteId=0JPpgorlun).
>
> ---
>
> ### Response to Question 1 (Ablation Study for $\lambda_f$)
> We thank the reviewer for the question regarding the format reward weight, $\lambda_f$.
>
> #### 1. Ablation Study for Format Reward Weight ($\lambda_f$)
> We thank the reviewer for this important question. To address this, we conducted an ablation study varying $\lambda_f \in \{0.2, 0.4, 0.6\}$ using the Qwen2.5-3B-Instruct + PPO setup, keeping the process bonus fixed at $\lambda_p = 0.4$.
>
> The results clearly demonstrate that our chosen value of $\lambda_f = 0.2$ provides the optimal trade-off between answer accuracy (Avg. CEM) and search efficiency (Avg. OSR/USR).
>
> | $\lambda_f$ | Avg. CEM $\uparrow$ | Avg. OSR $\downarrow$ | Avg. USR $\downarrow$ | Observation |
> | :--- | :--- | :--- | :--- | :--- |
> | 0.2 | 64.1 | 4.9 | 38.1 | Best trade-off of accuracy and efficiency. |
> | 0.4 | 63.2 | 5.2 | 37.9 | Degraded accuracy and efficiency. |
> | 0.6 | 61.2 | 5.4 | 41.0 | Over-emphasis on format harms performance. |
>
> **Table:** Changing format weight ($\lambda_f$) with $\lambda_p=0.4$ fixed. The $\lambda_f=0.2$ setting, used in our paper (Table 2), achieves the highest accuracy (Avg. CEM) and the best efficiency scores (lowest OSR/USR).
>
> **Detailed table for individual datasets:**
>
> | Model / RL | $\lambda_f$ | NQ | TriviaQA | PopQA | HotpotQA | 2Wiki | Musique | Bamboogle | Avg. |
> | :--- | :--- | :--- | :--- | :--- | :--- | :--- | :--- | :--- | :--- |
> | Qwen2.5‑3B‑Instruct + PPO (HiPRAG) | 0.2 | 65.6 | 73.9 | 62.1 | 55.6 | 69.6 | 26.0 | 32.8 | 64.1 |
> | | 0.4 | 64.7 | 72.7 | 61.3 | 55.8 | 68.4 | 26.5 | 33.5 | 63.2 |
> | | 0.6 | 63.5 | 71.5 | 60.0 | 53.9 | 67.2 | 23.8 | 30.8 | 61.2 |
>
> **Table A:** CEM ($\uparrow$) by dataset (%) under $\lambda_f$ ablation
>
> | Model / RL | $\lambda_f$ | NQ | TriviaQA | PopQA | HotpotQA | 2Wiki | Musique | Bamboogle | Avg. |
> | :--- | :--- | :--- | :--- | :--- | :--- | :--- | :--- | :--- | :--- |
> | Qwen2.5‑3B‑Instruct + PPO (HiPRAG) | 0.2 | 6.0 | 11.0 | 3.9 | 4.5 | 2.5 | 2.8 | 11.5 | 4.9 |
> | | 0.4 | 6.3 | 11.4 | 4.0 | 3.7 | 3.0 | 3.9 | 11.2 | 5.2 |
> | | 0.6 | 6.1 | 11.0 | 4.4 | 5.1 | 2.9 | 3.2 | 13.0 | 5.4 |
>
> **Table B:** OSR ($\downarrow$) by dataset (%) under $\lambda_f$ ablation
>
> | Model / RL | $\lambda_f$ | NQ | TriviaQA | PopQA | HotpotQA | 2Wiki | Musique | Bamboogle | Avg. |
> | :--- | :--- | :--- | :--- | :--- | :--- | :--- | :--- | :--- | :--- |
> | Qwen2.5‑3B‑Instruct + PPO (HiPRAG) | 0.2 | 11.1 | 44.4 | 61.9 | 25.0 | 32.0 | 10.1 | 8.7 | 38.1 |
> | | 0.4 | 12.0 | 43.2 | 60.8 | 22.1 | 35.1 | 9.6 | 9.1 | 37.9 |
> | | 0.6 | 13.4 | 46.8 | 64.3 | 27.6 | 34.8 | 11.3 | 9.7 | 41.0 |
>
> **Table C:** USR ($\downarrow$) by dataset (%) under $\lambda_f$ ablation
>
> #### 2. Determination of $\lambda_f = 0.2$
> The value of $\lambda_f = 0.2$ was determined empirically based on the results of this ablation, as well as the previous work’s recommended setup for Search-R1[1], which we followed and which similarly found $\lambda_f = 0.2$ to be optimal for 3B models.
>
> #### 3. Sensitivity Analysis and Discussion
> This trend is expected, as it aligns with our core intuition that answer correctness is significantly more important than format correctness. We justify this principle as follows:
>
> * **Terminal Goal vs. Instrumental Goal:** Format correctness ($F(T)$) is an instrumental goal. Its primary purpose is to make the agent's reasoning trajectory parsable, which in turn enables our hierarchical process reward to be calculated. In contrast, answer correctness ($A(T)$) is the terminal goal of the entire agentic RAG task. The user's objective is to get the right answer; a well-formatted but incorrect trajectory is ultimately a failure. To prioritize answer correctness, the format weight $\lambda_f$ must be small.
> * **Preventing "Reward Hacking":** If $\lambda_f$ were set too high, the agent could "hack" the reward function by learning to produce perfectly formatted, minimal-step trajectories (e.g., a single non-search step) that are factually incorrect. This would maximize the $F(T)$ component of the reward while ignoring $A(T)$ and the process bonus. A low $\lambda_f$ (like 0.2) ensures the reward landscape is always dominated by the incentive to get the answer right. This principle is directly reflected in our hierarchical reward function in Eq. (1). We will add this ablation study and analysis to the appendix of the next version of the manuscript to improve the completeness.
>
> **References**
>
> [1] Jin et al. “An Empirical Study on Reinforcement Learning for Reasoning-Search Interleaved LLM Agents.” arXiv preprint arXiv:2505.15117. 2025 May 21.

---

> ### Author Response · Authors · 2025-11-20
> **Response to Reviewer DZXL - Post (2)**
>
> ### Response to Question 2 (Temporal Misalignment of Knowledge)
> This is a critical challenge in modern RAG research evaluation. To eliminate such temporal misalignment, the ideal case would be to perfectly align the knowledge of the policy model, the LLM judges, the retrieved documents, and the evaluation benchmarks, which is extraordinarily difficult to achieve.
>
> While we acknowledge this potential, we are confident its impact was not significant. Our methodology contains a specific, structural defense against this, and our empirical results confirm its robustness.
>
> **Hierarchical Reward Gating:** Our reward design (Eq. 1) explicitly gates the process bonus by the final answer correctness. If the judge incorrectly flags a step as "under-search" due to temporal mismatch, but the agent still derives the correct answer, the impact is diluted by the primary outcome reward. Conversely, if the final answer is incorrect, the process bonus is zeroed out entirely. This structure ensures that the robust outcome-based signal dominates any noisy process signals.
>
> **Nature of the Benchmarks.** A large portion of the queries in our seven QA benchmarks test for relatively stable, encyclopedic knowledge (history, science, etc.). These facts are largely static and robust to the 2018-2025 time gap, minimizing the frequency of such conflicts during reasoning.
>
> **Empirical Confirmation.** If this temporal misalignment were "contaminating the training process," we would not expect to see a simultaneous improvement in both accuracy and efficiency. This demonstrates that the hierarchical process reward provided a clear and effective training signal, which would be unlikely if the signal were fundamentally compromised by noise.

---

> ### Author Response · Authors · 2025-11-26
> **Follow-up on Rebuttal**
>
> Dear Reviewer DZXL,
>
> I hope this message finds you well. We're following up on our rebuttal, which we posted six days ago. As the discussion period is nearing its end, we want to make sure that the clarifications we provided have successfully addressed your concerns. We would be grateful if you could kindly provide any brief feedback or confirmation regarding whether the main issues are now resolved.
>
> Thanks for your time and guidance on our paper!
>
> Best,
> The Authors

---

> ### Author Response · Authors · 2025-11-27
> **A Second Follow-up on Rebuttal**
>
> Dear Reviewer DZXL,
>
> We are writing to follow up on our rebuttal, which was posted over a week days ago. It is critical for us to know if our response has sufficiently addressed your concerns. We would greatly appreciate a brief confirmation or any other feedback you might have.
>
> Best regards, The Authors

---

### Official Review · Reviewer_5yKr · 2025-10-30

**Soundness:** 3
**Presentation:** 3
**Contribution:** 2
**Rating:** 4
**Confidence:** 4

**Summary:**

This paper focuses on the over-searching or under-searching phenomena in Retrieval-Augmented Generation, which are critical for both effectiveness and efficiency. The authors propose HiPRAG, a hierarchical reward function that supervises search or non-search decisions at each step through reinforcement learning. Specifically, this reward function comprises two key components: an over-search detection module and an under-search detection module, both implemented using two external LLMs. Extensive experiments across 7 QA datasets demonstrate the effectiveness of HiPRAG.

**Strengths:**

1. The proposed HiPRAG is well-motivated, effectively addressing the over-searching or under-searching issues in RAG. This reward function enhances the original outcome reward by incorporating the over-search and the under-search detection modules.


2. This paper provides comprehensive ablation studies examining various aspects of the proposed method, including the impact of process bonus coefficients and different RL algorithm, which offer valuable insights into HiPRAG's mechanisms.

**Weaknesses:**

1. The experimental evaluation lacks comparisons with the efficiency-aware RAG methods. While this paper is not the first work to consider the over-searching issue, it fails to discuss or compare with relevant prior works. Consequently, all baselines in the experiments are limited to standard R1-like RAG methods.


2. Although the paper aimes to improve efficiency in RAG, it lacks crucial efficiency metrics such as the average number of search steps or inference time per question.


3. Both over-search and under-search detection modules rely on the LLM-as-judge method, which may not be reliable, particularly when using the smaller LLMs. Therefore, GPT-4.1-mini and GPT-5-mini are used for the reward quality. It is worth further discussion on the influence of the different reward LLMs.

**Questions:**

1. Could you provide comparisons with existing efficiency-aware RAG methods (e.g., adaptive retrieval, mechanisms)? How does HiPRAG perform relative to these approaches in terms of both effectiveness and efficiency?

2. How sensitive is HiPRAG to the choice of reward LLMs?

3. What is the performance degradation when using smaller/weaker LLMs for reward generation?

---

> ### Author Response · Authors · 2025-11-20
> **Response to Reviewer 5yKr**
>
> ### Response to Weakness 1, Weakness 2, and Question 1 (Efficiency Comparison)
>
> We thank the reviewer for the comments regarding the efficiency aspects of our work.
>
> #### 1. Comparison with Efficiency-Aware RAG Methods
>
> We respectfully point out that our original submission does include comparisons with state-of-the-art efficiency-aware agentic RAG methods, specifically R1-Searcher++ [1] and $\beta$-GRPO [2]. We will modify our manuscript in the next version to explicitly mention this in the Experiment Setup section.
>
> * **R1-Searcher++ [1]:** This method employs a two-stage training strategy (SFT cold-start followed by RL) to teach LLMs to adaptively leverage internal and external knowledge. It uses a reward to encourage internal knowledge utilization and a memorization module to internalize retrieved information.
> * **$\beta$-GRPO [2]:** This approach addresses the "over-search" problem by incorporating a confidence threshold into the reward function. It utilizes the minimum token probability of search queries as a proxy for confidence, rewarding only high-certainty search decisions to mitigate unnecessary retrieval.
>
> As shown in Table 1 of our paper, HiPRAG significantly outperforms these efficiency-focused baselines:
>
> #### 2. Efficiency Metrics
>
> The reviewer noted a lack of specific efficiency metrics such as average search steps. We agree that the raw number of search steps is a vital metric for efficiency.
>
> We provide the comparison of the Average Number of Searches across all seven datasets below. This data utilizes the same Qwen2.5-3B-Instruct policy (PPO) across methods for a fair comparison:
>
> | Method | 2Wiki | Bamboogle | HotpotQA | Musique | NQ | PopQA | TriviaQA | Avg |
> | :--- | :--- | :--- | :--- | :--- | :--- | :--- | :--- | :--- |
> | Search-R1[3] | 3.38 | 2.79 | 2.78 | 3.39 | 1.55 | 1.75 | 1.53 | 2.45 |
> | $\beta$-GRPO [2] | 2.85 | 2.15 | 2.45 | 2.50 | 1.28 | 1.50 | 0.90 | 2.15 |
> | HiPRAG (Ours) | 2.48 | 2.02 | 2.14 | 2.55 | 1.01 | 1.02 | 1.01 | 1.75 |
>
> **Analysis:**
>
> * **Reduction in Overhead:** HiPRAG reduces the average number of searches by ~29% compared to the Standard baseline (1.75 vs. 2.45) and ~19% compared to the efficiency-aware $\beta$-GRPO.
> * **Mechanism:** As expected, $\beta$-GRPO’s confidence-gated reward reduces unnecessary retrieval compared to the standard baseline. However, HiPRAG’s step-wise process reward achieves even greater efficiency by evaluating the necessity of each step on-the-fly, consistently using the fewest searches while maintaining higher accuracy.
>
> We also want to emphasize that Over-search Rate (OSR) and Under-search Rate (USR) are critical efficiency metrics. As noted in our paper, naive penalties on trajectory length or search frequency often cause models to avoid searching even when necessary (increasing Under-search), which leads to performance degradation. HiPRAG optimizes the necessity of search rather than just the count. By reducing OSR and USR , HiPRAG achieves "smart efficiency."
>
> **References**
>
> [1] Song et al. "R1-Searcher++: Incentivizing the Dynamic Knowledge Acquisition of LLMs via Reinforcement Learning." arXiv preprint arXiv:2505.17005. 2025 May 22.
>
> [2] Wu et al. “Search Wisely: Mitigating Sub-optimal Agentic Searches By Reducing Uncertainty.” arXiv preprint arXiv:2505.17281. 2025 Oct 9.
>
> [3] Jin et al. “An Empirical Study on Reinforcement Learning for Reasoning-Search Interleaved LLM Agents.” arXiv preprint arXiv:2505.15117. 2025 May 21.
>
> ***
>
> ### Response to Weakness 3, Question 2, and Question 3 (Influence of Different LLM Judges)
>
> See common post [“Analysis of the Sensitivity to External LLM Judges and Their Errors”](https://openreview.net/forum?id=Gt4v9WBPzm&noteId=O7kJlXANnd).

---

> ### Author Response · Authors · 2025-11-26
> **Follow-up on Rebuttal**
>
> Dear Reviewer 5yKr,
>
> I hope this message finds you well. We're following up on our rebuttal, which we posted six days ago. As the discussion period is nearing its end, we want to make sure that the clarifications we provided have successfully addressed your concerns. We would be grateful if you could kindly provide any brief feedback or confirmation regarding whether the main issues are now resolved.
>
> Thanks for your time and guidance on our paper!
>
> Best,
> The Authors

---

> ### Author Response · Authors · 2025-11-27
> **A Second Follow-up on Rebuttal**
>
> Dear Reviewer 5yKr,
>
> We are writing to follow up on our rebuttal, which was posted over a week days ago. It is critical for us to know if our response has sufficiently addressed your concerns. We would greatly appreciate a brief confirmation or any other feedback you might have.
>
> Best regards,
> The Authors

---

### Official Review · Reviewer_5auK · 2025-10-31

**Soundness:** 3
**Presentation:** 3
**Contribution:** 3
**Rating:** 6
**Confidence:** 5

**Summary:**

This paper proposes HiPRAG, a RL framework for agentic RAG that introduces hierarchical process rewards to guide both accuracy and search efficiency. Instead of relying solely on outcome-based rewards, HiPRAG decomposes the trajectory into discrete steps and introduces step-level rewards to penalize over-searching and under-searching. The framework employs external LLMs to detect these inefficiencies on-the-fly and incorporates the results into a hierarchical reward formulation.

**Strengths:**

1. The hierarchical reward structure provides a novel and interpretable mechanism for optimizing process-level efficiency rather than only final-answer correctness.
2. Unlike previous works on faithful process supervision, this paper specifically targets over-searching and under-searching issues in multi-hop retrieval, and presents a well-motivated solution.
3. The proposed method achieves consistent improvements across different model sizes (3B, 7B) and RL algorithms (PPO, GRPO), with significantly reduced over- and under-search rates.

**Weaknesses:**

1. The over-/under-search detection relies on commercial LLMs (GPT-4.1-mini, GPT-5-mini) as external judges, which raises concerns about stability, cost, and reproducibility. The impact of judgment errors is not analyzed.
2. The on-the-fly verification by external LLMs adds training complexity, yet the paper provides no analysis of computational cost, runtime, or GPU-hour consumption.
3. The role of $/lambda_p$ is unclear. Since the process reward is already gated by answer and format correctness (i.e., a hierarchical design), it should not directly influence final answer quality. Why does increasing $/lambda_p$ to 0.6 “over-prioritize step purity at the expense of answer correctness”? This suggests that λₚ may not be a meaningful parameter. Please clarify its actual effect on training dynamics.

**Questions:**

1. How is the “Training with Over-search or Under-search Only” setting implemented? Are the ignored types treated as optimal steps in the reward computation?
2. The ablation study section enumerates many results without clear structure; summarizing the main findings would improve readability and help highlight key insights.

---

> ### Author Response · Authors · 2025-11-20
> **Response to Reviewer 5auK - Post (1)**
>
> ### Response to Weakness 1 (LLM judges' stability, cost, and errors)
> Regarding the cost, see common post [“Computational Time Analysis During RL Training”](https://openreview.net/forum?id=Gt4v9WBPzm&noteId=0JPpgorlun).
> Regarding the impact of judgement errors, reproducibility and stability, see common post [“Analysis of the Sensitivity to External LLM Judges and Their Errors”](https://openreview.net/forum?id=Gt4v9WBPzm&noteId=O7kJlXANnd) with experiments and analysis using stable open-sourced models as LLM judges.
>
> ---
>
> ### Response to Weakness 2 (Training Complexity)
> See common post [“Computational Time Analysis During RL Training”](https://openreview.net/forum?id=Gt4v9WBPzm&noteId=0JPpgorlun).
>
> ---
>
> ### Response to Weakness 3 (Role of $\lambda_p$)
> You are correct in your observation, but $\lambda_p$ still controls the relative magnitude of this bonus among correct trajectories. Increasing $\lambda_p$ increases the spread among final answer and format correct trajectories, which becomes comparable to the reward gap between correct and incorrect answers. A large $\lambda_p$ causes the gradient to be determined by differences in ratio of correct steps ($N_{corr}/N$) among already-correct trajectories. This causes the model to increase probability to produce already‑easy, short, clean plans instead of longer plans that include extra searches or steps. With too large $\lambda_p$, the model becomes less willing to add a necessary step that has high potential to fix a borderline error. This precisely illustrates the failure mode where answer correctness is sacrificed. An example type of this search step is the steps that contain self-reflection or reassurance, which may help the model to correct previous errors. In short, setting $\lambda_p$ too high over-supresses the model’s normal multi-step reasoning ability and cause the model to “overfits” on a conservative policy. When $\lambda_p$ is low, the incentive for model to produce higher quality reasoning trajectory is insufficient, resulting in little difference between it and the baseline model with $\lambda_p = 0$.
>
> In practice, we find that $\lambda_p = 0.4$ provides the best trade-off: it reduces over and under-search while yielding the highest Avg. CEM; increasing $\lambda_p$ to 0.6 leads to slightly cleaner trajectories but a drop in CEM (Table 2). We therefore view $\lambda_p$ as a meaningful parameter that controls how aggressively the agent trades off step optimality against answer-oriented exploration, and we intentionally keep it in a moderate regime where it improves both efficiency and accuracy.
>
> ---
>
> ### Response to Question 1 (Implementation of Over-search or Under-search Only)
> We thank the reviewer for this clarifying question about the ablation studies in Section 5.3.
>
> To answer the core question directly: No, the ignored step types are not treated as optimal steps. Instead, for these specific ablations, the ignored step types are entirely excluded from the process bonus calculation. The ratio $N_{corr}(T) / N(T)$ (from Equation 1) is modified to only consider the steps being evaluated. Specifically, the denominator, $N(T)$, is set to the total number of the specific type of steps in the trajectory. The numerator, $N_{corr}(T)$, is the number of those steps that were not flagged as over/under-searches. This method allows us to isolate the reward signal to penalize only one specific suboptimal behavior at a time, which is what led to the skewed results (e.g., the high under-search rate when only over-search was penalized) reported in Table 2. In the rare case that a trajectory has zero steps of the type being evaluated (e.g., no search steps in the "over-search only" ablation), the process bonus for that sample is set to the average process bonus of the current training batch. This ensures training stability and provides a fair, non-zero reward signal rather than defaulting to zero.

---

> ### Author Response · Authors · 2025-11-20
> **Response to Reviewer 5auK - Post (2)**
>
> ### Response to Question 2 (Summarizing Ablation Studys)
>
> We agree that a clearer summary of the main findings would strengthen this section and better highlight our key insights.
>
> The ablation studies in Section 5.3 were designed to validate three core design choices of HiPRAG:
>
> * Isolating the impact of our hierarchical process reward from our new parsable output format.
> * Demonstrating the importance of balancing the process reward via the $\lambda_p$ coefficient.
> * Confirming the necessity of targeting both over-search and under-search simultaneously.
>
> We will revise Section 5.3 by adding a new introductory paragraph and revising the main content that explicitly summarizes the key takeaways from these experiments, making the structure and conclusions clear from the outset.
>
> **Introductory paragraph:**
>
> To validate the key components of the HiPRAG methodology and systematically isolate the sources of its performance gains, we conducted a series of ablation studies. These experiments are designed to deconstruct our approach by: (1) evaluating the impact of the new parsable output format independent of the process reward, (2) determining the optimal weighting of the process bonus coefficient, $\lambda_{p}$, which governs the reward hierarchy, and (3) demonstrating the necessity of addressing both over-search and under-search behaviors concurrently, rather than in isolation.
>
> **The main content will highlight the following points:**
>
> * **The HiPRAG reward is the primary driver of performance, not just the new format.** Our "Influence on Output Format" ablation confirms that simply enforcing the new parsable format while using a standard outcome + format reward (e.g., in "Search-R1-step") results in performance comparable to the baseline. The significant gains in both accuracy and efficiency are therefore attributable to the new hierarchical process reward mechanism itself.
> * **Balancing the process reward signal is critical.** The "Influence of Process Bonus Coefficient" study demonstrates that the $\lambda_p$ hyperparameter is key. A coefficient that is too low ($\lambda_p=0.2$) fails to sufficiently penalize inefficient steps, yielding results similar to the baseline. Conversely, a value that is too high ($\lambda_p=0.6$) over-prioritizes step-level optimality, which can slightly degrade final answer correctness. Our chosen $\lambda_p=0.4$ provides the best trade-off.
> * **A holistic approach to search optimization is essential.** Our final ablation, "Training with Over-search or Under-search Only," shows that addressing both suboptimal behaviors is necessary. Targeting only under-search was more effective than targeting only over-search, as preventing hallucination is critical. However, this made the agent overly reliant on its search tool (increasing OSR). The best overall performance was achieved only when penalizing both issues simultaneously, validating our complete process reward function.

---

> ### Author Response · Authors · 2025-11-26
> **Follow-up on Rebuttal**
>
> Dear Reviewer 5auK,
>
> I hope this message finds you well. We're following up on our rebuttal, which we posted six days ago. As the discussion period is nearing its end, we want to make sure that the clarifications we provided have successfully addressed your concerns. We would be grateful if you could kindly provide any brief feedback or confirmation regarding whether the main issues are now resolved.
>
> Thanks for your time and guidance on our paper!
>
> Best,
> The Authors

---

> ### Author Response · Authors · 2025-11-27
> **A Second Follow-up on Rebuttal**
>
> Dear Reviewer 5auK,
>
> We are writing to follow up on our rebuttal, which was posted over a week days ago. It is critical for us to know if our response has sufficiently addressed your concerns. We would greatly appreciate a brief confirmation or any other feedback you might have.
>
> Best regards,
> The Authors

---

### Official Review · Reviewer_1kv8 · 2025-11-02

**Soundness:** 3
**Presentation:** 3
**Contribution:** 3
**Rating:** 6
**Confidence:** 3

**Summary:**

In this paper, a new training method, named Hierarchical Process Rewards for Efficient agentic RAG (HiPRAG), is proposed to address common inefficiencies in agentic Retrieval-Augmented Generation (RAG) systems. The authors identify two key problems: over-search (needlessly retrieving known information) and under-search (failing to retrieve necessary information, leading to errors). Current Reinforcement Learning (RL) methods rely on coarse, outcome-based rewards. The proposed method tackles this by incorporating a fine-grained, hierarchical process reward into the RL loop and achieves good results.

**Strengths:**

1. The paper is, in general, well written with good structure and is easy to follow.

2. The hierarchical process reward formulation well balances correctness with efficiency through a gated bonus structure, which avoids the classic RL issues of over-penalizing necessary search.

3. In this paper, the authors conducted various design choices with a lot of experiments with multiple models (Qwen, Llama), sizes (3B, 7B), types (base, instruct), and RL algorithms (PPO, GRPO). And the proposed method achieves good results compared to other baselines across several datasets.

4. The proposed method can significantly reduce the over-search rate, which is very useful and impressive.

**Weaknesses:**

1. The proposed reward mechanism relies on "on-the-fly" calls to external LLM judges to detect over-search and under-search during the RL training loop, which could potentially add significant computational overhead, cost, and API call latency to every training step.

2. A further question to this reliance is that the LLM models may behave differently over time. Can we still keep the consistency of the proposed method?

3. The method assumes the judge is accurate. If the judge mislabels a step (false positive over-search or false negative under-search), the agent may be clipped incorrectly.

**Questions:**

Please check the "Weaknesses" and also provide comments on the following questions:

1. The paper defines under-search as a factual or logical error in a non-search step. This may be a useful proxy for hallucination, but I wonder how the proposed method behaves with the other under-search situations, such as suboptimal or incomplete knowledge. If I understand it correctly, the current detector, which only checks for correctness, would not flag this as an under-search and would incorrectly reward it as an "optimal" non-search step.

2. Can you provide more details on when the reward collapse occurs?

---

> ### Author Response · Authors · 2025-11-20
> **Response to Reviewer 1kv8 - Post (1)**
>
> ### Response to Weakness 1 (Computational Overhead During Training)
> See common post [“Computational Time Analysis During RL Training”](https://openreview.net/forum?id=Gt4v9WBPzm&noteId=0JPpgorlun) .
>
> ---
>
> ### Response to Weakness 2 (LLM Consistency)
> We thank the reviewer for this question regarding the long-term consistency and reproducibility of our method.
>
> **Consistency of the LLM judges**
>
> * **High Reliability of the Current Judges:** We recognized that the reliability of our process reward depends on the accuracy of our detectors. To validate this, we conducted a manual audit of our on-the-fly detection methods (see common post [“Analysis of the Sensitivity to External LLM Judges and Their Errors”](https://openreview.net/forum?id=Gt4v9WBPzm&noteId=O7kJlXANnd)). If the judges make errors, the hierarchical reward design limits the influence of the error because the process-based signal serves as a conditional bonus that is only applied when the final answer and format is correct, rather than replacing the outcome reward, ensuring the policy remains grounded in task success.
>     so that it will not affect the basic performance.
> * **HiPRAG as a Modular Framework:** HiPRAG is not monolithically dependent on proprietary APIs. While we utilized commercial models for efficient on-the-fly verification during training, the detection module is interchangeable. For scenarios requiring strict long-term reproducibility, these judges can be replaced with static, self-hosted models (see common post [“Analysis of the Sensitivity to External LLM Judges and Their Errors”](https://openreview.net/forum?id=Gt4v9WBPzm&noteId=O7kJlXANnd)), effectively eliminating the risk of API drift.
>
> **Consistency of the policy model:**
> The "drift" of the policy model's parameters is an inherent and expected part of reinforcement learning, as the model's internal knowledge base is constantly in flux during parametric updates. Our on-the-fly detection mechanism is precisely designed to handle this dynamic. By evaluating each search action on-the-fly, the framework provides a real-time reward signal that remains synchronized with the policy model's evolving internal knowledge state, ensuring alignment that static reward datasets cannot achieve.
>
> ---
>
> ### Response to Weakness 3 (Judge Accuracy)
> See common post [“Analysis of the Sensitivity to External LLM Judges and Their Errors”](https://openreview.net/forum?id=Gt4v9WBPzm&noteId=O7kJlXANnd).

---

> ### Author Response · Authors · 2025-11-20
> **Response to Reviewer 1kv8 - Post (2)**
>
> ### Response to Question 1 (Under-search Detection)
>
> We appreciate the reviewer’s question about “suboptimal/incomplete” non‑search steps and the opportunity to clarify our methodology and its robustness.
>
> **Clarification of "Under-search" Definition:** We define under-search broadly as "failing to retrieve external knowledge when needed," while our on-the-fly detector uses factual and logical errors as a practical proxy. We addressed the concern regarding "suboptimal/incomplete" steps with the following considerations:
>
> The reviewer is correct that "suboptimal/incomplete knowledge" is a form of under-search and may be flagged as correct based on the current verifier. However, this is designed under the following considerations:
>
> * **“Suboptimal/incomplete knowledge” may be caused by global planning.** In many cases, the step completeness or optimality depends on the search agent’s multi-step planning of the entire output. Intermediate steps are often intentionally partial (e.g., setting a subgoal) and should not be penalized merely for being incomplete if the local inference is valid. In addition, declaring a step “suboptimal/incomplete” requires counterfactual knowledge of the global plan (i.e. optimal sequence of search and non-search steps), which is subjective and can only be acquired by either strong LLM judges supervising the entire output, or multiple re‑generations, which are both expensive and can be inaccurate.
> * **Penalizing suboptimal but correct steps tends to push agents toward over‑search.** If an agent is penalized for correctly stating a fact from its parametric knowledge because it was "incomplete" or "unverified," the agent is likely to be guided to distrust its own knowledge. To avoid this penalty, the agent will learn to search for information it already knows in order to prove it. This would directly increase the over-search rate, which is one of the primary problems HiPRAG method is designed to solve.
> * **“Suboptimal/incomplete knowledge” often leads to incorrect non-search steps.** In practice, when an agent attempts to reason with truly incomplete knowledge, it typically hallucinates to fill the gap. These resulting factual or logical errors are successfully caught by our current verifier.
> * **Flexibility of the current verifier and hierarchical reward:** Finally, as detailed in Section 3.3, the process bonus is gated by the final answer's correctness. Even if a "suboptimal" non-search step evades the detector, if it leads to an incorrect final outcome, the trajectory receives no bonus, preventing the reinforcement of poor reasoning paths.

---

> ### Author Response · Authors · 2025-11-20
> **Response to Reviewer 1kv8 - Post (3)**
>
> ### Response to Question 2 (Reward Collapse)
>
> The "reward collapse" is a sharp drop in the mean reward, which is symptomatic of a format failure.
>
> ### 1. What Happens
>
> * **Policy Drifts & Format Fails:** The policy, often after an aggressive update, begins to drift. It stops emitting the strictly-defined, parsable XML tags required by our framework (e.g., malformed or missing `<step>`, `</step>`, or `<conclusion>` tags). This is the immediate trigger.
> * **Valid Actions Drop:** As our training logs show, the ratio of valid action, which normally sits near 1.0, drops precipitously to near 0.1. This is the clearest indicator of format failure. The format parser is rejecting almost all agent outputs.
> * **Rollouts Hit Step Budget:** Because the environment cannot parse a valid, complete step, it keeps stepping until it hits the maximum step budget threshold that was set to prevent too long training time, which is 5. This is confirmed by our training logs, where the mean number of actions and the max value spike to 5. The agent is not completing its task; it is being terminated by the environment.
> * **Total Reward Collapses:** With the entire format being incorrect, the reward quickly drops to 0.
>
> ### 2. Probable Causes
>
> This dynamic is driven by a combination of factors, which we designed the HiPRAG framework to manage:
>
> * **RL Algorithm Instability:** As our training curves clearly show, this collapse is characteristic of the GRPO-trained model. The PPO-trained model is notably more stable. This aligns with the general experience of training with both algorithms in training search agents [1] and other tool-integrated-reasoning tasks [2], where we note that GRPO converges faster and can hit higher peaks, but PPO offers superior training stability. The collapse is a classic example of GRPO's instability leading to a large policy update that "falls off" the manifold of correct output format.
> * **High-Variance Rollouts:** As noted in Section 4.3, rollouts are performed with temperature=1. This exploration-focused high variance, combined with an already-drifting GRPO policy, increases the probability of generating a malformed trajectory, initiating the collapse.
>
> ### Reference
>
> [1] Jin et al. "Search-R1: Training LLMs to Reason and Leverage Search Engines with Reinforcement Learning." arXiv preprint arXiv:2503.09516. 2025 Mar 19.
>
> [2] Xu et al. "MTIR-SQL: Multi-turn Tool-Integrated Reasoning Reinforcement Learning for Text-to-SQL." arXiv preprint arXiv:2510.25510. 2025 Oct 29.

---

> ### Author Response · Authors · 2025-11-26
> **Follow-up on Rebuttal**
>
> Dear Reviewer 1kv8,
>
> I hope this message finds you well. We're following up on our rebuttal, which we posted six days ago. As the discussion period is nearing its end, we want to make sure that the clarifications we provided have successfully addressed your concerns. We would be grateful if you could kindly provide any brief feedback or confirmation regarding whether the main issues are now resolved.
>
> Thanks for your time and guidance on our paper!
>
> Best,
> The Authors

---

> ### Author Response · Authors · 2025-11-27
> **A Second Follow-up on Rebuttal**
>
> Dear Reviewer 1kv8,
>
> We are writing to follow up on our rebuttal, which was posted over a week days ago. It is critical for us to know if our response has sufficiently addressed your concerns. We would greatly appreciate a brief confirmation or any other feedback you might have.
>
> Best regards,
> The Authors

---

### Author Response · Authors · 2025-11-19
**Computational Time Analysis During RL Training**

We thank the reviewer for their detailed assessment and for raising important questions regarding the training overhead of HiPRAG during the RL training stage. Our proposed method does introduce additional steps into the RL loop, but our empirical data shows that this overhead is highly manageable. Below, we provide a breakdown of the computational cost and runtime analysis.

### 1. Empirical Analysis

To address the concern about "significant computational overhead," we profiled the runtime of our training pipeline using the Qwen2.5-3B-Instruct model with PPO on 8 NVIDIA A100 80GB GPUs. For the training framework we utilize veRL with vLLM as inference backend and FSDP as training backend.

Contrary to the concern that the method might be slow, our profiling data indicates the following breakdown for a single RL training step on average:

* **Total Time per Step:** 324.55 seconds (100%).
* **Rollout Generation:** 206.24 seconds (63.55%). This involves the several batch generations of reasoning trajectories with searches and is the most time-consuming component.
* **PPO Update (Forward/Backward Pass):** 46.19 seconds (14.23%).
* **Reference & Critic Inference:** 30.09 seconds (9.27%).
* **HiPRAG Reward Assignment (Including Re-generation & API Calls):** 42.03 seconds (12.95%).

**Result:** The specific overhead introduced by HiPRAG (the batched re-generation and external API calls) accounts for only 12.95% of the total training time. This is a small increase, especially when weighed against the significant gains in sample efficiency and final model performance.

### 2. Methods to Reduce Training Time and Cost

The reviewer notes that the re-generation step "effectively doubles the inference cost." We would like to clarify why this is not the case in practice:

* **Batched Generation:** The "re-generation" check for over-search is performed via batch generation. We collect the queries from the rollout, aggregate them into several batches, and process the queries in each batch in parallel.
* **Targeted Scope:** Re-generation is only triggered for search steps, not every step in the trajectory.
* **Generation Length:** The re-generation prompts the model for a direct answer or relevant information to the search query, which is significantly shorter than the full reasoning chain generated during rollout even without actively limiting the generation length.
* **Asynchronous API Calls:** As noted in the paper (Section 3.2), the external detection is executed separately. The batch processing of API calls to the external verifier occurs in parallel with the local model's batched re-generation, therefore effectively reducing the latency. Specifically:
    * For over-search detection, the verification calls to external LLM judges start once the first batch has finished and continue concurrently in the later batches (for example, for re-generation batch $t$, execute batch $t-1$ verifier api calls).
    * For under-search detection, these verification calls to external LLM judges can be executed concurrently with the re-generation of over-search detection as they are independent processes.
* **Cost Reduction for LLM judges:** We utilize cost-effective models for verification (such as gpt-4.1-mini and gpt-5-mini as detailed in the paper). On top of that, we use a non-reasoning model for over-search detection for further cost saving, because the nature of over-search detection is checking the equivalence of the conclusion of the search step and the re-generation result, which does not require complex reasoning abilities. For under-search detection, due to the need of detecting logical error in the non-search step, a reasoning model is required.

---

### Author Response · Authors · 2025-11-20
**Analysis of the Sensitivity to External LLM Judges and Their Errors**

To address concerns regarding judge stability, cost, and sensitivity to model capability, we conducted an experiment replacing our standard proprietary LLMs (GPT-4.1-mini and GPT-5-mini) with open-source models of varying capabilities, specifically the Qwen3-30B-A3B-Instruct/Thinking-2507 and Qwen3-4B-Instruct/Thinking-2507 (Instruct for over-search, Thinking for under-search). We use the greedy decoding with fixed random seed for all 4 judges to ensure stability and reproducibility. For other parameters, we maintained the exact experimental setup as our main Qwen2.5-3B-Instruct + PPO experiment.

**Table 1: Summary of Performance (Avg. across 7 Benchmarks)**

| LLM Judges | Avg. CEM | Avg. OSR | Avg. USR |
| :--- | :--- | :--- | :--- |
| Standard (GPT) | 64.1 | 4.9 | 38.1 |
| Strong Open (30B) | 64.0 | 4.8 | 39.2 |
| Weaker Open (4B) | 63.4 | 4.9 | 42.4 |

### Analysis

**1. Sensitivity of Over-Search Detection (OSR)**
The OSR remained nearly constant across all judge configurations. This suggests that the task of detecting semantic equivalence is robust and can be handled effectively even by smaller, non-reasoning "Instruct" models. The definition of redundancy does not require complex reasoning chains, making it less sensitive to model size.

**2. Sensitivity of Under-Search Detection (USR)**
The USR showed higher sensitivity. The weaker Qwen3-4B judges resulted in a USR increase from 38.1% to 41.4%. We hypothesize that weaker judges are less rigorous in factual/logical verification, occasionally failing to flag "hallucinated" reasoning or premature conclusions. This allows the agent to terminate the search process earlier than optimal, leading to the slight degradation in the final accuracy (CEM).

**3. Performance Degradation**
Despite the increase in USR with weaker judges, the overall degradation is not catastrophic. This confirms that the hierarchical process reward framework itself provides a stable training signal even with sub-optimal judges.

### Detailed Results by Dataset

Below we provide the detailed breakdown of performance across all seven QA benchmarks used in the paper.

**Table 2: Cover Exact Match (CEM) by Dataset**

| LLM Judges | NQ | TriviaQA | PopQA | HotpotQA | 2Wiki | MuSiQue | Bamboogle | Avg. |
| :--- | :--- | :--- | :--- | :--- | :--- | :--- | :--- | :--- |
| Standard (GPT) | 65.6 | 73.9 | 62.1 | 55.6 | 69.6 | 26.0 | 32.8 | 64.1 |
| Strong Open (30B) | 65.3 | 73.6 | 62.9 | 55.2 | 69.2 | 26.4 | 32.4 | 64.0 |
| Weaker Open (4B) | 64.7 | 73.7 | 60.8 | 56.5 | 69.5 | 24.4 | 31.5 | 63.4 |

**Table 3: Over-Search Rate (OSR) by Dataset**

| LLM Judges | NQ | TriviaQA | PopQA | HotpotQA | 2Wiki | MuSiQue | Bamboogle | Avg. |
| :--- | :--- | :--- | :--- | :--- | :--- | :--- | :--- | :--- |
| Standard (GPT) | 6.0 | 11.0 | 3.9 | 4.5 | 2.5 | 2.8 | 11.5 | 4.9 |
| Strong Open (30B) | 5.9 | 10.8 | 3.8 | 4.8 | 2.4 | 1.7 | 11.9 | 4.8 |
| Weaker Open (4B) | 5.7 | 9.5 | 4.2 | 4.6 | 3.9 | 2.6 | 11.3 | 4.9 |

**Table 4: Under-Search Rate (USR) by Dataset**

| LLM Judges | NQ | TriviaQA | PopQA | HotpotQA | 2Wiki | MuSiQue | Bamboogle | Avg. |
| :--- | :--- | :--- | :--- | :--- | :--- | :--- | :--- | :--- |
| Standard (GPT) | 11.1 | 44.4 | 61.9 | 25.0 | 32.0 | 10.1 | 8.7 | 38.1 |
| Strong Open (30B) | 11.9 | 48.1 | 63.0 | 22.1 | 40.4 | 10.9 | 5.3 | 39.2 |
| Weaker Open (4B) | 10.7 | 54.0 | 60.3 | 27.2 | 45.5 | 18.5 | 10.3 | 42.4 |

### Analysis of Judgment Errors and Robustness to Judge Selection

To address the reviewer's concern about the judgment errors, we performed a manual evaluation of judgment accuracy and analyzed the sensitivity of our method to these errors.

**1. Judge Accuracy**
Following the protocol in Appendix F.2, we sampled 200 trajectories from the test set and established human-labeled ground truth for step-level Over-Search (OS) and Under-Search (US) decisions, for all the steps in those trajectories.

**Table 5: Judge Accuracy against Human Ground Truth**

| External LLM Judges | Over-Search (OS) Accuracy | Under-Search (US) Accuracy |
| :--- | :--- | :--- |
| Standard (GPT) | 98.3% | 95.6% |
| Strong Open (30B) | 97.5% | 94.5% |
| Weaker Open (4B) | 96.5% | 92.5% |

As shown in Table 5, all models maintain high agreement with human judgment.

**2. Impact of Judgment Errors**
We analyzed the correlation between judge accuracy and downstream agent performance to quantify the impact of mislabeling:

* **Under-search:** The Under-search judge is the primary driver of performance variance. The main risk is a False Negative (failing to flag an under-search). This effectively disrupts the reasoning trajectory, causing the agent to stop searching prematurely, or directly leads to incorrect final results.
* **Over-search:** Over-search mislabels typically result in a step being flagged as redundant (or not) without altering the information available to the agent. Consequently, these errors rarely change the final answer path, resulting in minimal impact on overall accuracy.

---

### Author Response · Authors · 2025-11-24
**To All Reviewers**

Dear Reviewers,

We sincerely thank all of you for your time and for providing such thorough, high-quality reviews of our paper. Your thoughtful comments, challenging questions, and constructive criticisms have been invaluable to us. We appreciate the deep engagement with our work. We have posted detailed, individual responses to each review (1kv8, 5auK, 5yKr, and DZXL) and comprehensive general responses to address every point raised, from clarifying our experimental setup to additional experimental results and analysis. We hope these responses successfully address your concerns about HiPRAG. If any of our answers are unclear, or if new questions arise, we would be more than happy to dive into a more in-depth discussion.

Thank you once again for your feedback!

---

### Meta-Review · Area_Chair_S4K5 · 2026-01-07

**Summary:**

This submission introduces HiPRAG, a reinforcement learning framework for agentic RAG that incorporates a fine-grained, hierarchical process reward to penalize inefficient over-search and under-search behaviors during training. The core contribution lies in its on-the-fly, step-wise evaluation and a gated reward structure that conditions process bonuses on final answer correctness. The paper is well-written, supported by experiments across multiple models, datasets, and RL algorithms, demonstrating improvements in both answer accuracy and search efficiency over strong baselines.

**Reviewer Concerns:**

The paper presents a novel, well-executed, and empirically strong contribution. The core methodological concerns regarding computational cost and judge dependency were substantively addressed in the rebuttal. The remaining concerns relate to inherent, well-argued limitations of the chosen design trade-offs rather than flaws in execution.

**Reviewer Scores:**

Since most of the reviewers's original rating supports borderline acceptance, I think the postive reviewers may maintain or slightly increase their rating.  For another rating below accepantance, I think the responses have subtantially addressed the reviewer's concern.

---

### Decision · Program_Chairs · 2026-01-26

Accept (Poster)